

# Modeling apparent Pb loss in zircon U-Pb geochronology

Glenn R. Sharman[1], Matthew A. Malkowski[2]

[1]Department of Geosciences, University of Arkansas, Fayetteville, AR 72701, USA
[2]Department of Geological Sciences, Jackson School of Geosciences, University of Texas at Austin, Austin, TX 78712, USA

*Correspondence to*: Glenn R. Sharman (gsharman@uark.edu)

**Abstract.** Although the loss of radiogenic Pb from zircon is known to be a major factor that can cause inaccuracy in the U-Pb geochronological system, the distribution of Pb loss in natural samples has not been well characterized. Treatment of zircon by chemical abrasion (CA) has become standard practice in isotope dilution-thermal ionization mass spectrometry (ID-TIMS), but CA is much less commonly employed prior to *in-situ* analysis via laser ablation-inductively coupled plasma-mass

spectrometry (LA-ICP-MS) or secondary ionization mass spectrometry (SIMS). Differentiating the effects of low levels of Pb loss in Phanerozoic zircon with relatively low precision *in-situ* U-Pb dates, where the degree of Pb loss is insufficient to cause discernible discordance, is challenging. We show that U-Pb dates that have been perturbed by Pb loss may be modeled by convolving a Gaussian distribution, that represents the unperturbed U-Pb date distribution, with a distribution that characterizes Pb loss. We apply this mathematical framework to model the distribution of apparent Pb loss in 10 igneous samples that have

both non-CA LA-ICP-MS or SIMS U-Pb dates and an estimate of the crystallization age, either through CA U-Pb or [40]Ar/[39]Ar geochronology. All but one sample showed negative age offsets that were unlikely to have been drawn from an unperturbed U-Pb date distribution. Of the eight continuous distribution types we considered, modeling apparent Pb loss using the Weibull distribution produced, on average, the closest match with the non-CA U-Pb date distributions. We show two contrasting patterns in apparent Pb loss: samples where most zircon U-Pb dates undergo a bulk shift and samples where most zircon U-Pb

dates exhibited low age offset but fewer grains had more significant offset. Our modeling framework allows comparison of relative degrees of apparent Pb loss between samples of different age, with the first and second Wasserstein distances providing useful estimates of the total magnitude of apparent Pb loss. Given that the large majority of *in-situ* U-Pb dates are acquired without the CA treatment, this study highlights a pressing need for improved characterization of apparent Pb loss distributions in natural samples to aid in interpreting non-CA *in-situ* U-Pb data and to guide future data collection strategies.

## 1 Introduction

Zircon U-Pb geochronology is arguably one of the most important radiometric dating approaches used by geoscientists, with widespread application to constraining the age of Pleistocene and older geologic materials (Schoene, 2013; Gehrels, 2014). We rely on zircon U-Pb dates for calibrating the geological time scale (e.g., Compston, 2000a; 2000b; Bowring and Schmitz, 2003; Gradstein et al., 2004; Kaufmann, 2006), constraining the timing of important Earth history events (Froude et al., 1983;

Schoene et al., 2010; 2015; Burgess et al., 2014), and determining the rates of Earth processes (Rioux et al., 2012; Schoene et



al., 2012; Johnstone et al., 2019; Schoene et al., 2019). The zircon U-Pb geochronometer is particularly powerful due to the ability to assess agreement between the $^{238}U \rightarrow ^{206}Pb$ and $^{235}U \rightarrow ^{207}Pb$ decay chains, with $^{206}Pb^*/^{238}U$ and $^{207}Pb^*/^{235}U$ dates in agreement plotting on the Concordia line, where * indicates radiogenic Pb (Wetherill, 1956). For example, a zircon that has undergone loss of radiogenic Pb will be pulled off the Concordia line towards the origin, thus becoming discordant.


The causes and complications of open system behavior (e.g., radiogenic Pb loss) in zircon have long been a topic of concern (Tilton et al., 1955), with recognizing and mitigating the effects of Pb loss remaining a major challenge. For example, due to the shape of the $^{206}Pb^*/^{238}U$ vs $^{207}Pb^*/^{235}U$ Concordia line, Pb loss that occurs within several 100's Myr after crystallization results in discordance developing at a very low angle relative to the Concordia line. This 'sliding along concordia' effect can

make Pb loss difficult to discern, particularly in relatively low-precision *in-situ* (i.e., LA-ICP-MS or SIMS) datasets when the Pb loss only produces modest discordance (e.g., <10%; Bowring and Schmitz, 2003; Ireland and Williams, 2003; Reimink et al., 2016; Spencer et al., 2016; Watts et al., 2016; Anderson et al., 2019). Such low levels of Pb loss have been termed 'cryptic' and may be associated with spatial heterogeneities including radiation-damaged U-rich zones and microstructures (Nasdala et al., 2005; Kryza et al., 2012; Watts et al., 2016). Most Pb loss in zircon is likely a consequence of recrystallization or Pb

transport in crystals with severe radiation damage (Silver and Deutsch, 1963; Cherniak and Watson, 2001; Mezger and Krogstad, 2004; Marsellos and Garver, 2010). Mechanisms for Pb loss may include metamorphism (Kroner et al., 1994; Orejana et al., 2015; Zeh et al., 2016), hydrothermal alteration (Geisler et al., 2002, 2003); diagenetic fluids or fluid flow (Willner et al., 2003; Morris et al., 2015), and chemical weathering (Stern et al., 1966; Black, 1987; Balan et al., 2001; Pidgeon et al., 2017; Andersen and Elburg, 2022). Pb loss is thought to primarily occur at temperatures <250°C in which radiation

damage in zircon is unable to be annealed over geologic timescales (Schoene, 2013).

Zircon domains that have lost Pb may be preferentially removed by first thermally annealing the zircon at high temperature (e.g., 800-1100°C) and then partially dissolving the zircon in a heated HF solution in a technique called chemical abrasion (CA) (Mattinson, 2005). The CA treatment is now routinely applied in ID-TIMS analysis and has contributed to both improved

precision and accuracy of CA-ID-TIMS U-Pb data (Schoene, 2013). Although some *in-situ* U-Pb laboratories practice thermal annealing routinely (e.g., Allen and Campbell, 2012; Solari et al., 2015), CA has been applied much less frequently (Crowley et al., 2014; von Quadt et al., 2014; Watts et al., 2016; Ver Hoeve et al., 2018; Ruiz et al., 2022). Several studies that have conducted paired analysis of non-CA and CA of the same samples via *in-situ* U-Pb geochronology have found the non-CA U-Pb dates to skew younger than the CA U-Pb dates (Crowley et al., 2014; von Quadt et al., 2014; Watts et al., 2016). A growing

number of maximum depositional age studies with tandem non-CA LA-ICP-MS and CA-ID-TIMS dating have shown the youngest non-CA U-Pb dates to skew young relative to CA U-Pb dates or other geologic constraints, even when considering measurement uncertainty (e.g., Herriott et al., 2019; Schwartz et al., 2022; Howard et al., 2022). However, there is a lack of quantitative constraints on the relative importance of Pb loss in influencing non-CA U-Pb date distributions acquired via *in-situ* mass spectrometry, particularly as related to influencing depositional age constraints (Copeland, 2020).




This study presents a novel framework for quantifying the effects of Pb loss on untreated (i.e., non-CA) U-Pb date distributions. We first suggest that Pb loss-perturbed U-Pb date distributions may be viewed as the convolution of two signals: a Gaussian distribution that reflects the unperturbed U-Pb date distribution and the distribution that characterizes Pb loss. We then apply this mathematical framework to model the distribution of apparent Pb loss that has affected 10 igneous samples of Miocene to

Carboniferous age. Our results highlight the importance of quantifying apparent Pb loss to better understand the potential influence on non-CA U-Pb date distributions, with a need for improved characterization to better resolve both the distribution types and magnitudes associated with Pb loss in zircon.

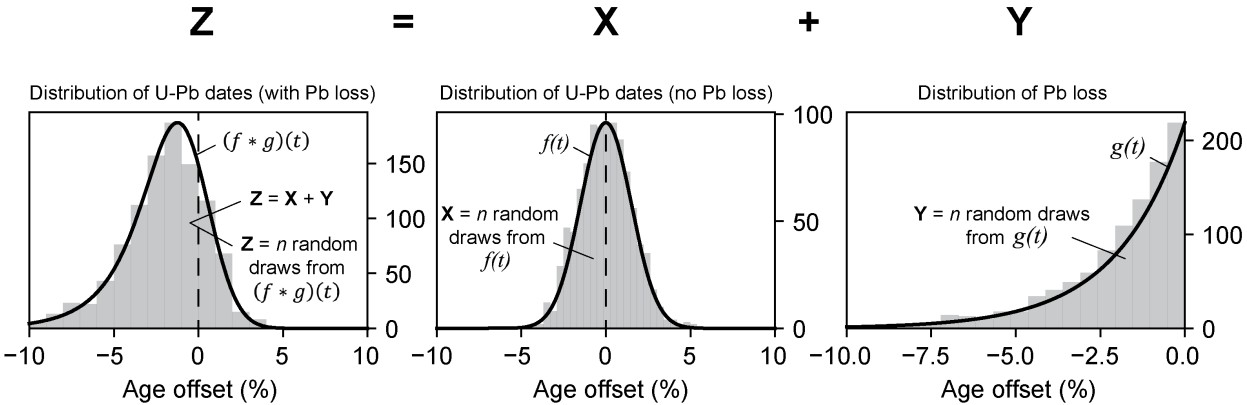

**Figure 1. Illustration of how *n* zircon U-Pb dates that have been perturbed by Pb loss (Z) may be modeled as the summation of *n* non-perturbed U-Pb dates (X) and the amount of Pb loss encountered by each (Y). X is drawn from *f(t)* that reflects the Gaussian distribution of U-Pb dates that are unperturbed by Pb loss and Y is drawn from *g(t)* that represents the distribution of Pb loss in the sample. The distribution that characterizes Z may be found by convolving *f(t)* and *g(t)*. Although we assume that *f(t)* is a Gaussian distribution, the distribution type of Pb loss, *g(t)*, shown in this example as an exponential function, could take a number of discrete or continuous forms (Figs. 2 and 3). Note that in our modeling framework, values of X, Y, and Z are normalized as percentage deviation from the true age (i.e., the mean of *f(t)*), where negative values indicate that the measured U-Pb date is younger than the true age. See Supplemental Video 1 for an animation that illustrates the process of convolution.**

## 2 Mathematical framework

A series of *n* measurements of Pb loss-perturbed U-Pb dates, **Z**, may be modeled as the sum of the corresponding

unperturbed U-Pb dates, **X**, and the amount of Pb loss encountered by each date, **Y**,

$$\mathbf{Z} = \mathbf{X} + \mathbf{Y} \qquad \qquad \text{(Equation 1)}$$

where **Z**, **X**, and **Y** are all 1-D matrices with *n* values (Fig. 1). Because Pb loss produces a lower U-Pb date, the values of **Y** must be negative in our formulation of Equation 1 (Fig. 1). We assume that unperturbed U-Pb dates (**X**) from cogenetic zircon may be characterized by a Gaussian distribution, *f(t)*, whose mean (μ) equals the crystallization age and standard

deviation (σ) reflects the degree to which the values of **X** deviate from the shared crystallization age. If these U-Pb dates are





then subjected to Pb loss (**Y**) that is drawn from *g(t)*, and if **X** and **Y** are independent, then **Z** may be viewed as being drawn from the convolution of *f(t)* and *g(t)*

$$(f * g)(t) = \int_{-\infty}^{\infty} f(\tau)g(t - \tau)d\tau \qquad \text{(Equation 2)}$$

where *g(t)* is reflected about the y-axis and shifted in $\tau$ space (Fig. 1; Supplemental Video 1). We model Pb loss as
percentage offset from the crystallization age rather than deviation in absolute time (i.e., Myr) to promote comparison of samples of different age (Fig. 1). Because Pb loss is always negative and at most -100% of the crystallization age, we constrain *g(t)* to values between 0% and -100%.

$$g(t) = \begin{cases} 0\% \; if \; g(t) > 0\% \\ g(t) \; if -100\% \leq g(t) \leq 0\% \\ -100\% \; if \; g(t) \leq -100\% \end{cases}$$

Equation 2 may be solved analytically for some forms of *f(t)* and *g(t)*. For example, the convolution of Gaussian and exponential distributions is known as the exponentially modified Gaussian distribution (Grushka, 1972: Fig. 1; Supplemental Video 1). However, $(f * g)(t)$ may also be solved numerically, which has the advantage of allowing both *f(t)* and *g(t)* to take any form. Figure 2 illustrates the effects of three discrete forms of *g(t)* on normally distributed U-Pb dates drawn from *f(t)*. A sample that experienced no Pb loss can be thought of as the convolution of *f(t)* with a discrete form of *g(t)* where the
100% of probability corresponds to 0% Pb loss (Fig. 2a). Having 100% of probability for a discrete amount of Pb loss that is >0% produces a bulk shift in the U-Pb age distribution (i.e., constant Pb loss; Fig. 2b). Similarly, Pb loss experienced by only a subset of grains, or isolated Pb loss, may also be modeled using a discrete distribution, (Fig. 2c). Alternatively, the Pb loss function *g(t)* could be represented by a continuous probability distribution, where values of Pb loss vary continuously between values of 0% and -100% (Fig. 3). Figure 3 illustrates the effects of eight different types of continuous distributions
on modifying a Gaussian distribution following convolution.

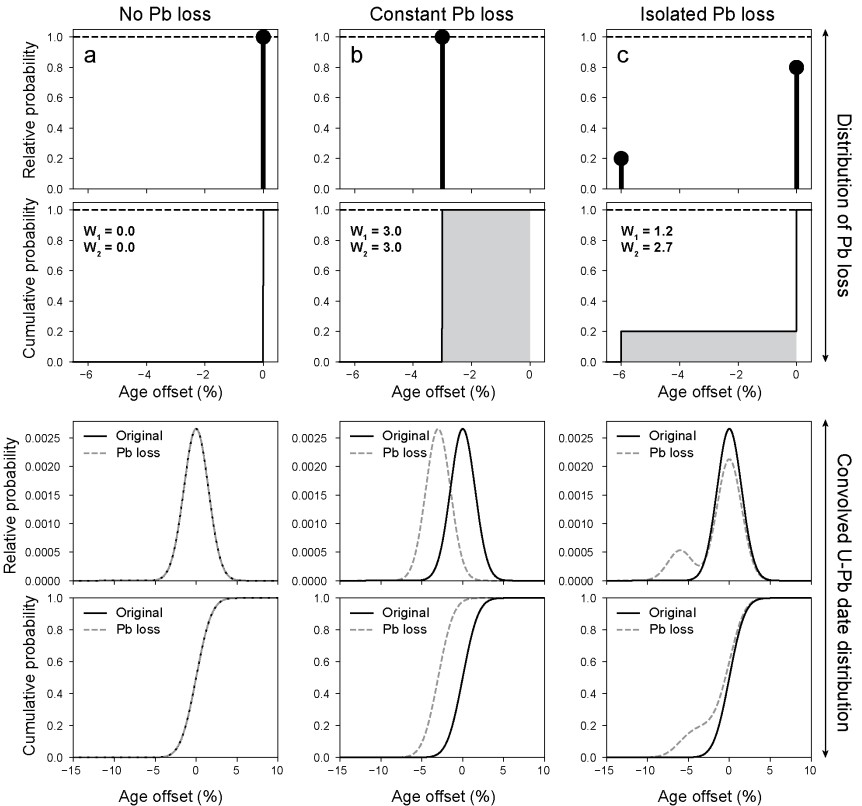

**Figure 2. Illustration of how normally distributed zircon U-Pb dates may be perturbed by discrete distributions of Pb loss. The top row represents the distribution of Pb loss in the sample expressed as a percentage of the true age, where the height of the black bar and ball indicates the relative probability of the specified age offset. Three discrete scenarios are shown: a) no Pb loss, b) constant Pb loss, and c) isolated Pb loss. The grey shading indicates the area beneath the Pb loss cumulative distribution function. The bottom row shows both the relative (above) and cumulative (below) probabilities of the unperturbed (solid black line) and Pb loss-perturbed (dashed grey line) U-Pb date distributions.**



**Figure 3. Illustration of how normally distributed U-Pb dates may be perturbed by continuous distributions of Pb loss (see Figure 2 caption). Scale and shape parameters follow the implementation of these distributions in the Python SciPy library (Virtanen et al., 2020).**



## 3 Methods

### 3.1 Modeling approach

We use the mathematical framework described above to model both the distribution of apparent Pb loss, *g(t)*, experienced by a group of cogenetic grains and their unperturbed U-Pb date distribution, *f(t)*. Because *g(t)* could represent any geological or analytical process that introduces negative age offsets, we use the phrase "apparent Pb loss" when describing our modeled estimates of *g(t)*. For instance, matrix-related systematic errors (Allen and Campbell, 2012), addition of U-Th during weathering (Pigdeon et al., 2019), and even sample contamination from younger minerals could introduce negative age shifts

exclusive of loss of radiogenic Pb.

To model *g(t)*, we allow the μ of *f(t)* to vary within the 95% confidence interval associated with an independent estimate of the crystallization age. We then estimate both *g(t)* and σ of *f(t)* by iteratively solving for the combination of parameters that minimize the misfit between the measured U-Pb dates and modeled distribution $(f * g)(t)$ using the Python

scipy.optimize.minimize() function. We define misfit as the sum of squared residuals between the empirical cumulative distribution function (ECDF) of the measured U-Pb dates and the cumulative density function (CDF) of the modeled U-Pb age distribution. In total, we consider 11 different distribution types for *g(t)* that consist of both discrete (no Pb loss, constant Pb loss, and isolated Pb loss; Fig. 2) and continuous (uniform, gamma, exponential, Rayleigh, Weibull, Pareto, half-normal, and lognormal; Fig. 3) distributions.


If both non-CA and CA analyses are available from the same sample, then the distribution of CA U-Pb dates may be used to constrain the parameters of *f(t)*. For such samples, we modify the approach described above by first finding the Gaussian distribution *f(t)* that most closely approximates the treated U-Pb date distribution. We then use this best-fitting *f(t)* in estimating *g(t)* using the minimization-of-misfit technique described above. Such datasets have the advantage of providing

constraints on σ of *f(t)*, which is otherwise treated as an unknown parameter during modeling if only non-CA U-Pb dates are available.

### 3.2 Samples

We apply the mathematical and modeling framework presented above to estimate the distribution of apparent Pb loss in 10 igneous samples that range in age from Carboniferous to Miocene, nine of which have been published previously (Table 1).

Samples CTU, RCP, and SRF are all from upper Eocene rhyolites of the Caetano caldera system of the western United States (Watts et al., 2016). These samples were split into non-CA and CA aliquots prior to analysis via SIMS (Watts et al., 2016). We used the error-weighted mean age of the CA U-Pb dates as an estimate of the true crystallization age for each sample, with



weighted means approximately 0.4-0.6 Myr older than the corresponding $^{40}Ar/^{39}Ar$ sanidine ages (Watts et al., 2016). The number of analyses per aliquot (non-CA or CA) ranges from 17-34 for these three samples (Table 1).


### Table 1. Sample Summary

| Sample | Age (Ma) | Reference | N (non-CA) | N (CA) | f(t) (Ma) | g(t) type | g(t) sum of squared residuals | g(t) parameters | g(t) P2.5-P50-P97.5 (%) | $W_1$ | $W_2$ |
|---|---|---|---|---|---|---|---|---|---|---|---|
| | | | | | | | | | Model results (best fit of continuous distributions) | | |
| ELM18 DVTC-10 | 15.7 ± 0.2 (2σ)[1] | Miller et al. (2022) | 144 | n.a. | 15.69 ± 0.69 (1σ) | Gamma | 0.97 | scale = 14.2 shape = 0.37 | -30.97 -2.07 -0.02 | 5.6 | 10.4 |
| 248-2 | 24.422 ± 0.25 (2σ)[3] | von Quadt et al. (2014) | 30 | 55 | 24.40 ± 0.55 (1σ) | Weibull | 3.20 | scale = 1.51 shape = 0.72 | -9.32 -0.94 -0.02 | 1.9 | 3.2 |
| 029-5[5] | 24.480 ± 0.084 (2σ)[3] | von Quadt et al. (2014) | 42 | 48 | 24.45 ± 0.79 (1σ) | Lognormal | 3.27 | scale = 4.21 shape = 0.46 | -10.38 -4.22 -1.71 | 4.7 | 5.2 |
| 059-1[5] | 24.57 ± 0.28 (2σ)[2] | von Quadt et al. (2014) | 41 | 36 | 24.4 5± 0.95 (1σ) | Lognormal | 1.08 | scale = 2.76 shape = 0.52 | -7.73 -2.77 -0.99 | 3.2 | 3.6 |
| CTU | 34.41 ± 0.26 (2σ)[2] | Watts et al. (2016) | 24 | 18 | 34.48 ± 0.68 (1σ) | Uniform | 0.93 | -6.79% (max) -0.45% (min) | -6.64 -3.62 -0.61 | 3.6 | 4.1 |
| RCP | 34.38 ± 0.32 (2σ)[2] | Watts et al. (2016) | 34 | 18 | 34.40 ± 0.69 (1σ) | Half normal | 2.96 | scale = 3.46 | -7.76 -2.34 -0.11 | 2.8 | 3.5 |
| SRF | 34.62 ± 0.37 (2σ)[2] | Watts et al. (2016) | 17 | 17 | 34.43 ± 0.73 (1σ) | Weibull | 1.36 | scale = 0.47 shape = 0.53 | -5.79 -0.30 -0.01 | 0.9 | 2.0 |
| DG 026 | 76.41 ± 0.45 (2σ)[3] | von Quadt et al. (2014) | 31 | 34 | 76.15 ± 1.40 (1σ) | Uniform | 2.09 | -5.36% (max) -0.01% (min) | -5.24 -2.7 -0.15 | 2.7 | 3.1 |
| MM20-EC-109[6] | 144.50 ± 0.07 (2σ)[4] | This study | 68 | n.a. | 144.43 ± 3.17 (1σ) | Pareto | 1.22 | shape = 0.98 | -28.98 -1.00 -0.03 | 3.8 | 9.9 |
| AvQ 244[7] | 333.60 ± 0.66 (2σ)[3] | von Quadt et al. (2014) | 17 | 19 | 333.61 ± 10.66 (1σ) | Half normal | 8.16 | scale = 9.64 | -21.62 -6.5 -0.3 | 7.7 | 9.6 |

[1]Sanidine $^{39}Ar/^{40}Ar$ age (Snow and Lux, 1999)
[2]Error-weighted mean of chemically abraded U-Pb dates
[3]Concordia age (CA-ID-TIMS)
[4]Error-weighted mean 5 of 5 zircon crystals analyzed via CA-ID-TIMS
[5]U-Pb dates older than 28 Ma excluded from analysis
[6]U-Pb dates older than 158 Ma excluded from analysis
[7]U-Pb dates older than 360 Ma excluded from analysis
N = Number of analyses
n.a. = Not available
$W_1$ = first Wasserstein distance
$W_2$ = second Wasserstein distance



We present analysis of five samples reported in von Quadt et al. (2014), including upper Oligocene andesite/trachy-andesite
from Macedonia (248-2, 029-5, and 059-1), upper Cretaceous dolerite from Romania (DG026), and middle Carboniferous
granite from West-Bulgaria (AvQ 244). These samples were also split into non-CA and CA aliquots prior to analysis via LA-
ICP-MS. For samples other than 059-1 we use concordia ages from CA-ID-TIMS analyses of between three and six crystals
for the crystallization age of each sample (von Quadt et al., 2014; Table 1). For sample 059-1 we used the weighted mean of
the CA U-Pb dates. The number of analyses per sample (non-CA or CA) ranged from 17-55 for this dataset (Table 1).


Sample ELM18DVTC-10 is from a Miocene ash-flow tuff from the Pangua Formation in the western United States that has
U-Pb dates acquired via LA-ICP-MS (Miller et al., 2022). We use a $^{40}Ar/^{39}Ar$ weighted mean age of $15.7 \pm 0.2$ Ma ($2\sigma$)
from the same unit as an estimate of the crystallization age of this sample (sample 592-GV1 of Snow and Lux, 1999). Sample
ELM18DVTC-10 was highlighted by Schwartz et al. (2022) who noted the youngest zircon U-Pb dates to be much younger
than the accepted $^{40}Ar/^{39}Ar$ age of this unit. Miller et al. (2022) also noted the presence of these young grains and suggested
that they may be a consequence of surface contamination from units higher in the section.

Sample MM20-EC-109 is a Lower Cretaceous intermediate ash interbedded within marine carbonaceous mudstone from the
Rio Mayer Formation of Argentina with 68 zircon U-Pb dates acquired via LA-ICP-MS (Table A2). Laser ablation spot
locations were selected on the rim and/or core of the grain guided by CL images (Figure A2), with 59 grains in total analyzed.
We use a crystallization age of $144.43 \pm 0.07$ Ma ($2\sigma$) derived from a weighted mean average of five zircon crystals analyzed
via CA-ID-TIMS at the Boise State University Isotope Geology Laboratory (Table A3). This sample exhibits a large offset
between the youngest U-Pb dates acquired via LA-ICP-MS, up to ~60% younger than the CA-ID-TIMS weighted mean.

### 3.3 Statistical analysis

To evaluate the likelihood that the measured U-Pb date distribution could have been drawn from the modeled $(f * g)(t)$, we
apply the nonparametric, 1-sided Kolgomorov-Smirnov (K-S) and Kuiper statistical tests that compare the ECDF with the
cumulative CDF of $(f * g)(t)$ (Press, 2007). The Kuiper statistic is relatively more sensitive in differences in the tails of the
distributions versus the K-S statistic (Vermeesch, 2018). We reject the null hypothesis that the non-CA U-Pb dates were drawn
from $(f * g)(t)$ if the K-S or Kuiper p-value is <0.05 (i.e., 95% confidence level). We thus interpret p-values >0.05 to indicate
that the non-CA U-Pb dates could have been plausibly drawn from $(f * g)(t)$ at a 95% confidence level (Press, 2007).
However, it should be noted that Saylor and Sundell (2016) found that both K-S and Kuiper p-values more frequently reject
the null hypothesis than expected. We thus use p-values as a general guideline to model goodness-of-fit.

The Wasserstein distance has been recently proposed as a metric for quantifying the dissimilarity between detrital zircon U-
Pb age distributions (Lipp and Vermeesch, *in review*). We consider the first and second Wasserstein distances, $W_1$ and $W_2$, to
be useful approximations for the total degree of negative age perturbation that a set of analyses has experienced,



$$W_1 = \int_0^1 |M^{-1} - N^{-1}| dt \qquad \text{(Equation 3)}$$

$$W_2 = \sqrt{\int_0^1 |M^{-1} - N^{-1}|^2 dt} \qquad \text{(Equation 4)}$$

where $M^{-1}$ and $N^{-1}$ are the inverses of the CDFs M and N. Because values of Pb loss are restricted to between -100% and 0%,

both $W_1$ and $W_2$ yield maximum possible values of 100 (i.e., 100% of grains have -100% age offset, or the U-Pb system is completely reset). The $W_1$ simply equates to the area beneath the cumulative probability distribution of the apparent Pb loss function (e.g., Fig. 3). Because the $W_2$ distance involves a squaring of the distance between the quantile functions, it imparts a higher cost penalty for the part of the distribution with strongly negative offset values. For example, the $W_1$ and $W_2$ distances are equal for a Pb loss function characterized by constant Pb loss (e.g., -3% Pb loss produces $W_1$ and $W_2$ values of 3, Fig. 2).

However, the $W_2$ distance is often much larger than $W_1$ for Pb loss distributions with a heavy tail, such as the Pareto distribution (Fig. 3). As such, the $W_2/W_1$ ratio provides an approximation of Pb loss distribution asymmetry, with values of 1 indicating constant Pb loss and values >>1 indicating highly asymmetric Pb loss.

## 4 Results

Figure 4 presents a summary of model misfit for each of the 10 samples and 11 distribution types considered. Of the three discrete Pb loss distributions considered, isolated Pb loss yielded the lowest average misfit with a value 4.07 (Fig. 4).

The isolated Pb loss scenario also produced the closest match with samples RCP, SRF, and DG026. The scenario of no Pb loss, however, performed the worst of any scenario that we considered, with an average misfit of 97.4 (Fig. 4). Correspondingly,

both K-S and Kuiper p-values for the no Pb loss

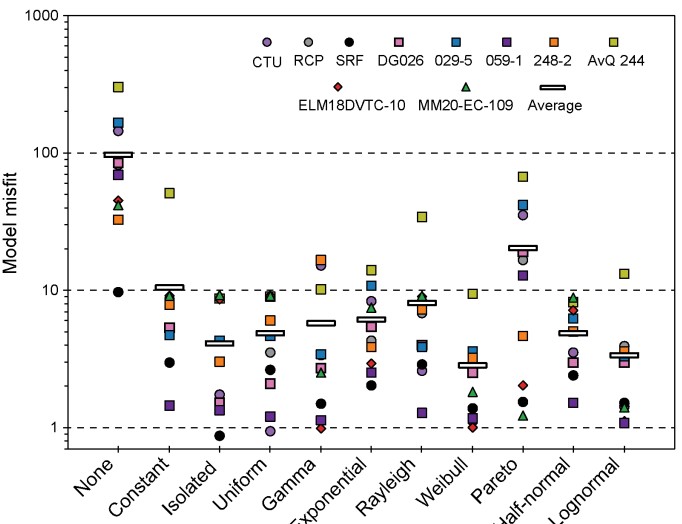

**Figure 4.** Summary of model misfit for each of the 11 distribution types and 10 samples modeled (see Table A1 for modeling results).





**Figure 5. Modeling of apparent Pb loss in zircon U-Pb dates acquired via LA-ICP-MS or SIMS. The best-fitting continuous distribution of apparent Pb loss is shown (Table 1; see Figure A1 for plots for all samples and apparent Pb loss distribution types). Ranked date plots with non-CA (red) and CA (blue) U-Pb analyses are shown with 2σ uncertainties. Empirical cumulative distribution functions (ECDFs) are shown as solid lines while model results are shown as dashed lines. See text for further discussion of model results.**





scenario are <<0.05 for all samples except SRF, suggesting that the untreated LA-ICP-MS or SIMS U-Pb dates are unlikely to have been drawn from an unperturbed U-Pb date distribution. Of the continuous distributions considered, the Weibull and lognormal distributions produced the overall best fits, with average misfit values equaling 2.82 and 3.33, respectively (Fig. 4). The Pareto distribution produces a heavy-tail (Fig. 3) that yielded good fits for some samples with extreme outlying values (e.g., ELM18DVTC-10 and MM20-EC-109) but poor fits for some of the other samples (average misfit of 20.1, Fig. 4). The other distribution types yielded intermediate results with average misfit values ranging between 4.8 and 8.1 (Fig. 4). With a few exceptions, p-values for both the K-S and Kuiper tests are >0.05 for the continuous distribution types we modeled (Table A1).

Figure 5 presents a comparison of actual vs modeled U-Pb date distributions for each sample, with the best-fitting continuous apparent Pb loss distribution shown (Table 1; see Figure A1 for individual plots that show the fit for each sample and distribution type). We chose to not consider discrete distributions of apparent Pb loss for the "best" fit because we consider it unlikely that Pb loss (or other processes that cause negative age offsets) would be limited to discrete values (e.g., Fig. 2). The best-fitting continuous distribution types include the gamma (ELM18DVTC-10), Weibull (248-2 and SRF), lognormal (029-5 and 059-1), uniform (CTU), half normal (RCP), and Pareto (MM20-EC-109) distributions (Table 1). $W_1$ distances ranged between 0.9 (sample SRF) and 7.7 (sample AvQ 244) and $W_2$ distances between 2.0 and 10.4 (Table 1; Fig. 5). In general, the Pareto, gamma, Weibull, and lognormal distributions are more likely to predict more abundant extreme values of age offset than the Rayleigh or uniform distributions (Fig. 5, Table 1, Fig. A2).

To further examine variations in the distributions of apparent Pb loss between samples, we plotted the best-fit Weibull distributions (Fig. 6). Even though the Weibull distribution was not the closest match for each sample, it yielded the best overall matches across samples, with misfit values <10 for all samples (Fig. 4). Figure 6 displays two distinct behaviors of apparent Pb loss when modeled using the Weibull distribution. Samples with a Weibull shape parameter <1 have their maximum relative probability of apparent Pb loss close to 0% suggesting a strongly decaying rate of age offset (i.e., most Pb loss-perturbed U-Pb dates have very little age offset, while a few have more significant offset). These samples also displayed $W_2/W_1 \geq 1.7$. Samples with a Weibull shape parameter >1, however, display a tendency for the mode of apparent Pb loss to be >0%, representing more of a bulk shift in age (e.g., most U-Pb dates have some age offset, while relatively few have very little or very much age offset) that produced $W_2/W_1 \leq 1.3$.



## 5 Discussion

### 5.1 Assumptions and limitations

The mathematical and modeling framework that we present includes several underlying assumptions and limitations that should be considered.

1.      When modeling apparent Pb loss via convolution we make the assumption that the amount of age offset is uncorrelated with crystallization age. We consider this to be a reasonable assumption for zircon crystals of similar age and with a shared history (e.g., most igneous samples). However, the relative amount of Pb loss experienced by zircon of different ages could be variable. Older zircon have experienced more radioactive decay and thus are more prone to radiation damage (Marsellos and Garver, 2010). Similarly, some zircon incorporate elevated concentrations (>1000 ppm) of U into their crystal structure, which promotes accelerated radiation damage and degrades the zircon matrix (White and Ireland, 2012). In particular, Pb loss is believed to be most common at temperatures below ~250°C, at which radiation damage cannot be healed (Schoene, 2013). Thus, detrital samples with zircon of widely varying age and provenance are more likely than igneous samples to violate the assumption that Pb loss is independent of age. We consider some strategies for analyzing detrital zircon below in Section 5.3.

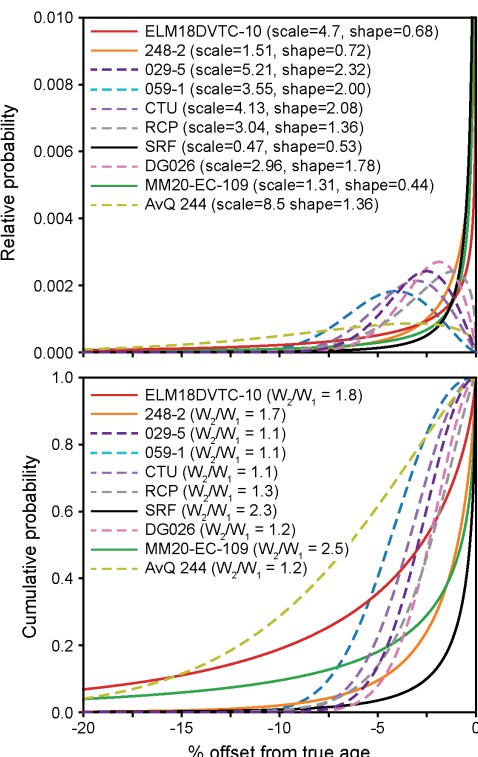

**Figure 6. Distributions of apparent Pb loss when modeled as a Weibull distribution. Samples where the Weibull shape parameter is >1 are shown as a dashed line.**

2.      For datasets with paired non-CA and CA measurements, our modeling approach assumes that the relative precision of the analyses is similar. This is because the Gaussian distribution that best approximates the CA U-Pb date distribution, $f(t)$, is convolved with the apparent Pb loss distribution $g(t)$ to fit the non-CA U-Pb date distribution. The Watts et al. (2016) SIMS dataset shows similar relative precision regardless of treatment approach (non-CA vs CA). Some samples from the von Quadt et al. (2014) LA-ICP-MS dataset exhibit slightly lower relative precisions for non-CA versus CA, with sample AvQ 244 yielding the largest difference with an average relative precision of 1.1% (1σ) for non-CA dates and 0.8% (1σ) for CA dates. We suggest that for the purposes of modeling apparent Pb loss, paired non-CA and CA U-Pb datasets should be collected on the same instrument using similar acquisition parameters to avoid introducing large changes in measurement precision. Alternatively, the CA U-Pb dates may be used to only constrain the μ of f(t) in the model, with σ treated as an unknown parameter (e.g., for paired non-CA LA-ICP-MS and CA-ID-TIMS datasets; Figs. 5a and 5i).





3.        For datasets with paired non-CA and CA measurements, we do not take into account any imperfections of the
chemical abrasion process. For example, although the CA treatment aims to completely remove all radiation damaged zones
of the crystal (Mattinson, 2005), it is possible to have remaining residual zones of Pb loss following treatment (e.g., Schoene
et al., 2010). Any such remaining compromised domains of the crystal will yield at least some apparent Pb loss when analyzed.
For instance, Watts et al. (2016) interpreted three zircon U-Pb analyses from SRF to have some residual Pb loss that was not

fully accounted for by the CA process. Incorporation of Pb loss-perturbed U-Pb dates when modeling $f(t)$ would likely produce
an underestimate of the true magnitude of the apparent Pb loss. Additionally, the CA process may itself damage grains to the
point of being unrecoverable for future steps (e.g., mounting, analysis). Because loss of grains via CA is likely to correlate
with age, geochemistry, and/or history (e.g., older, highly metamict, or recycled grains), there is a potential for differences
between the non-CA and CA U-Pb date distributions that do not relate to apparent Pb loss. Analyzing a dataset with a metamict-

prone age fraction could result in such differences being mistakenly interpreted as the result of apparent Pb loss if a particular
group of zircon are selectively removed by CA. We suspect that this issue may be more prevalent in detrital samples that
exhibit greater diversity in zircon characteristics than in igneous samples.

### 5.2 Distributions of apparent Pb loss

What distribution type(s) characterize apparent Pb loss in natural samples? Our results strongly suggest that at least nine of the

10 samples modeled have at least some systematic negative age offset that cannot be explained by random measurement
uncertainties alone. This is because the K-S and Kuiper statistical tests are unable to reject the null hypothesis for many of the
apparent Pb loss distribution types considered (Table A1). For example, only the no Pb loss scenario produced a p-value <0.05
for sample MM20-EC-109, suggesting that any of the other 10 modeled distributions of apparent Pb loss may be statistically
plausible for this sample. With the exception of ELM18DVTC-10 which has 144 non-CA LA-ICP-MS analyses, the samples

we analyzed have relatively low numbers of analyses (between 17 and 68, average of 32) for a given sample and treatment
category (non-CA or CA) (Table 1). We hypothesize that collection of larger-$n$ datasets would allow better differentiation
between possible apparent Pb loss distribution types, particularly because different distributions can produce similarly looking
functions (Fig. 3). In some cases, different distribution types can produce identical probability density functions (e.g., the
Weibull distribution interpolates between the exponential and Raleigh distributions).

Even if the specific distribution type that characterizes apparent Pb loss cannot be uniquely identified, our analysis suggests
two contrasting behaviors in apparent Pb loss (Fig. 6). We speculate that U-Pb dates that undergo a bulk shift (i.e., $W_2/W_1 \cong$
1) may reflect a population of zircon crystals with relatively homogenous characteristics (e.g., size, U content, etc.) that have
all experienced a similar post-crystallization history. Correspondingly, the population of zircon that produces U-Pb dates with

a highly asymmetric distribution of age offset (i.e., $W_2/W_1 > \sim$1.5) may reflect heterogeneity between crystals, with variable
characteristics and/or post-crystallization histories. Collection of size measurements and trace element concentrations from





zircon in addition to measurement of the U-Pb date (e.g., Watts et al., 2016), would likely help evaluate hypotheses about the underlying factors that control such behavior of apparent Pb loss distributions. Furthermore, given the relatively small number of samples modeled in this study, we suggest that there is a need for more samples to undergo paired non-CA and CA characterization to improve understanding of the range of behaviors that may be typical. For example, it is presently unclear whether it is more common for samples to have their U-Pb dates bulk shifted (e.g., samples 029-5, 059-1, CTU, DG026) versus having relatively few U-Pb dates highly offset (e.g., samples MM20-EC-109 and ELM18DVTC-10; Fig. 5).

Why do some samples experience more overall apparent Pb loss than others? Although we anticipated that apparent Pb loss would be greater for older samples, our analysis shows no clear trend by sample age (although we acknowledge that the relatively high degree of apparent Pb loss modeled in the youngest sample, ELM18DVTC-10, may be a consequence of contamination from overlying units, instead of true Pb loss; Miller et al., 2022). Even the three samples from the same Eocene caldera system (CTU, RCP, and SRF) showed contrasting amounts of apparent Pb loss ($W_2$ ranges from 2.0 to 4.1; Table 1) as noted by Watts et al. (2016). Characterizing the overall magnitude of apparent Pb loss in a wider range of samples would likely help elucidate predictive factors, if any.

### 5.3 Detrital and other multi-modal samples

The modeling framework presented above is designed for a group of cogenetic crystals with a shared crystallization age (e.g., autocrystic zircon from the same magmatic episode; Miller et al., 2007). This requirement stems from our definition of apparent Pb loss as a relative shift, or percentage deviation from the crystallization age (Fig. 1). Because detrital samples are typically multi-modal (i.e., zircon are derived from >1 underlying Gaussian distribution), we cannot usually assume that the measured U-Pb dates were all drawn from the same Gaussian distribution, $f(t)$. For example, volcanic arcs crystallize zircon over the span of 10's of millions of years through multiple cycles of magmatism (Paterson and Ducea 2015). Thus, a detrital zircon age spectra with a volcanic arc source may appear as a broad distribution of U-Pb dates that are themselves multimodal or yield distribution tails representing waning magmatic fluxes (Caricchi et al., 2014). Failure to recognize the true heterogeneity in crystallization age in such a sample could cause an incorrect interpretation of the apparent Pb loss distribution.

However, a sample with zircon crystals derived from multiple Gaussian distributions (e.g., detrital samples or igneous samples with antecrystic, xenocrytic, or inherited zircon; e.g., Rossignol et al., 2019) may also be modeled using this approach provided that the unperturbed U-Pb date distribution (e.g., CA-LA-ICP-MS or CA-SIMS) can be unmixed into its constituent Gaussian distributions (e.g., Sambridge and Compston, 1994). We illustrate this approach in Figure 7 where synthetic U-Pb dates are drawn from a mixture of three Gaussian distributions and convolved with the same Weibull distribution of apparent Pb loss. The resulting Pb loss-perturbed U-Pb date distribution (e.g., non-CA LA-IC-MS or SIMS) can be produced by separately convolving each Gaussian distribution with the Pb loss distribution, $g(t)$, multiplying by the appropriate weighting factor, and then summing them up to unity (Fig. 7b). By modeling 1,000 synthetic U-Pb dates drawn from both the unperturbed and





perturbed distributions, we found a modeled *g(t)* (scale = 4.6, shape = 1.3) that was a close match for the true *g(t)* (scale = 5, shape = 1.5) (Fig. 7c). In practice, one could apply this approach by conducting Gaussian mixture modeling on a CA-LA-ICP-MS or CA-SIMS dataset and then convolving each Gaussian separately with the apparent Pb loss distribution when minimizing the misfit.

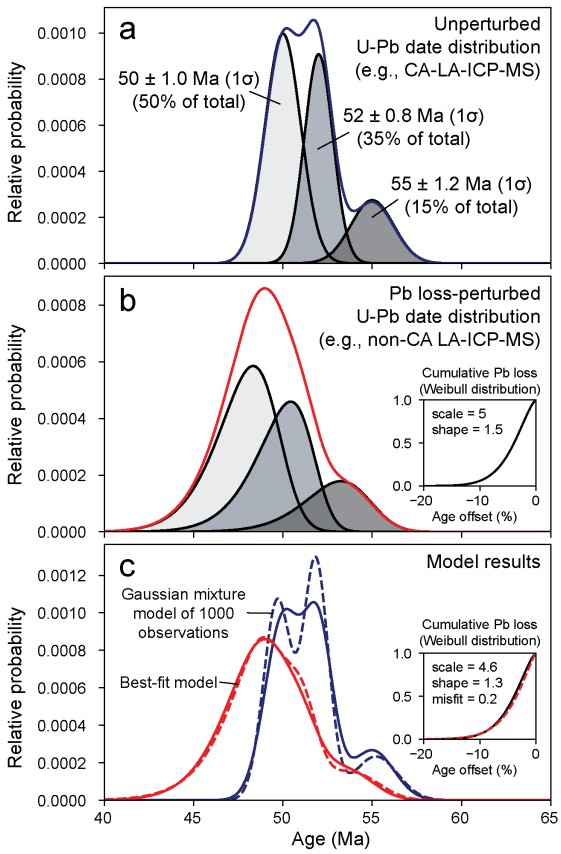

**Figure 7. Illustration of how a model of apparent Pb loss could be applied to U-Pb age distributions drawn from multiple Gaussian distributions (e.g., detrital samples). a) Three Gaussian distributions with different means and standard deviations are mixed together at proportions of 50-35-15 to form the solid blue line. The solid blue line could represent a detrital U-Pb date distribution that is unperturbed by Pb loss (e.g., as measured by CA-LA-ICP-MS). b) The three Gaussian distributions in (a) are each convolved with a Weibull distribution (scale = 5, shape = 1.5) that represents the distribution of Pb loss. The solid red line represents the mixture of these modified Gaussian distributions at the same proportions as in (a). The solid red line could represent a detrital U-Pb date distribution that has been perturbed by Pb loss (e.g., as measured by non-CA LA-ICP-MS). Note how age peaks in (a) become blurred and younger in (b). c) The dashed blue line shows the results of Gaussian mixture modeling of 1,000 synthetic observations drawn randomly from the solid blue line. Note that while the dashed blue line does not exactly fall on the solid blue line, it represents a reasonable approximation of the underlying distribution. The dashed red line shows the best-fit result when the three modeled (unmixed) Gaussian distributions are convolved with a Weibull distribution. Note that the shape and scale values of the modeled Pb loss distribution (dashed red line; 4.6 and 1.3, respectively) are close to the true values of 5 and 1.5 shown as the solid black line.**

One word of caution with this approach is that our modeling framework assumes that the degree of age offset is independent of the age of the sample (see also point #1 in Section 5.1). To counteract this potential issue, we suggest that a U-Pb date distribution with age modes that span a significant reach of time be broken up into groups and modeled separately. In this way, the apparent Pb loss distribution that characterizes a young age mode would be allowed to vary from the apparent Pb loss distribution that has affected an older age mode.

## 5.4 Importance of quantifying the distribution of apparent Pb loss in *in-situ* U-Pb geochronology


The overwhelming majority of published *in-situ* U-Pb dates from zircon, minimally >600,000 and likely in the millions of analyses (Puetz et al., 2021), have not been treated using CA. In contrast, CA is now practiced routinely in the ID-TIMS



community which has contributed to growing precision and accuracy over the past two decades (Schoene, 2013). However, the strategy of mitigating Pb loss through avoidance is perhaps less easily adopted to routine *in-situ* U-Pb geochronology. For

instance, there may be practical limitations with chemically abrading large numbers of grains, including the potential loss of certain age modes that would be detrimental to provenance analysis. We thus suggest that there is a pressing need to improve quantitative characterization of apparent Pb loss distributions in non-CA *in-situ* U-Pb datasets to aid in interpreting these datasets and to guide strategies for future data collection.

It is somewhat concerning that nine of the 10 samples analyzed in this study exhibited statistically significant amounts of negative age offset from the estimated true crystallization age. Even a small age offset of a few percent, or cryptic Pb loss (Kryza et al., 2012; Watts et al., 2016), has potentially important repercussions for interpreting the age and rates of geologic events and processes. For example, there is a growing awareness in the detrital geochronological community that the youngest zircon U-Pb dates often skew unexpectedly young relative to the plausible crystallization age (e.g., Herriot et al., 2019; Gehrels

et al., 2020; Schwartz et al., 2022). Presently, there is no consensus on the importance of post-depositional Pb loss on influencing depositional age interpretations (e.g., Herriott et al., 2019; Copeland, 2020; Schwartz et al., 2022). Sample MM20-EC-109 illustrates the risk well; we initially interpreted the young tail on the U-Pb date distribution to suggest a depositional age of ~125 Ma based on the youngest cluster of overlapping U-Pb dates. The youngest single analysis was a $60.5 \pm 2.4$ Ma rim on a $135.3 \pm 3.0$ Ma core, with the second youngest being a $79 \pm 1.2$ Ma date measured from the core of a grain, with the

corresponding rim yielding an older $129.8 \pm 3.6$ Ma date (Table A2). Interpretation of the youngest single U-Pb date or dates as the depositional age of this sample would have produced a highly erroneous estimate, off by up to -58% of the true eruption age of $144.50 \pm 0.07$ ($2\sigma$) Ma as determined by CA-ID-TIMS. Because this ash is interbedded within a sequence of organic rich marine mudstone in the Austral Basin of Argentina, the misinterpretation in this case could have led to an erroneous depositional age model with implications for interpreting the paleoclimatic and geodynamic context of these sediments.


Although modeling detrital samples was outside of the scope of this study, we believe that our results bear upon maximum depositional age analysis. The tendency for the youngest U-Pb dates in a sample to be affected by Pb loss (or other similar process) complicates even conservative estimates of the maximum depositional age (Dickinson and Gehrels., 2009; Coutts et al., 2019; Schwartz et al., 2022). If apparent Pb loss follows a continuous distribution (e.g., Fig. 3), then it is ill-advised to





assume that outlying U-Pb dates may be rejected while the rest are considered unperturbed (see also discussion in Copeland, 2020). Even an interpretation based on the peak age probability of the youngest age mode is

likely to be too young, because the process of convolution produces a young-shift in the mode of the distribution, in addition to creating a young tail (Figs. 3 and 7). Because existing methods of calculating the maximum

depositional age (Dickinson and Gehrels, 2009; Coutts et al., 2019; Vermeesch, 2021) fail to account for systematic negative age offsets, our analysis suggests that there is a high probability for erroneous estimates of the maximum

depositional age if (1) there are a large number of zircon crystals with crystallization ages that are close to the age of deposition, (2) the overall number of measured U-Pb analyses is very high, and/or (3) the magnitude of apparent Pb

loss is high. In addition, the distribution type of apparent Pb loss (e.g., heavy-tailed vs not) will also influence maximum depositional age calculations due to varying probability of finding extremely offset values, with samples

with large $W_2$ values particularly at risk of highly inaccurate estimates.

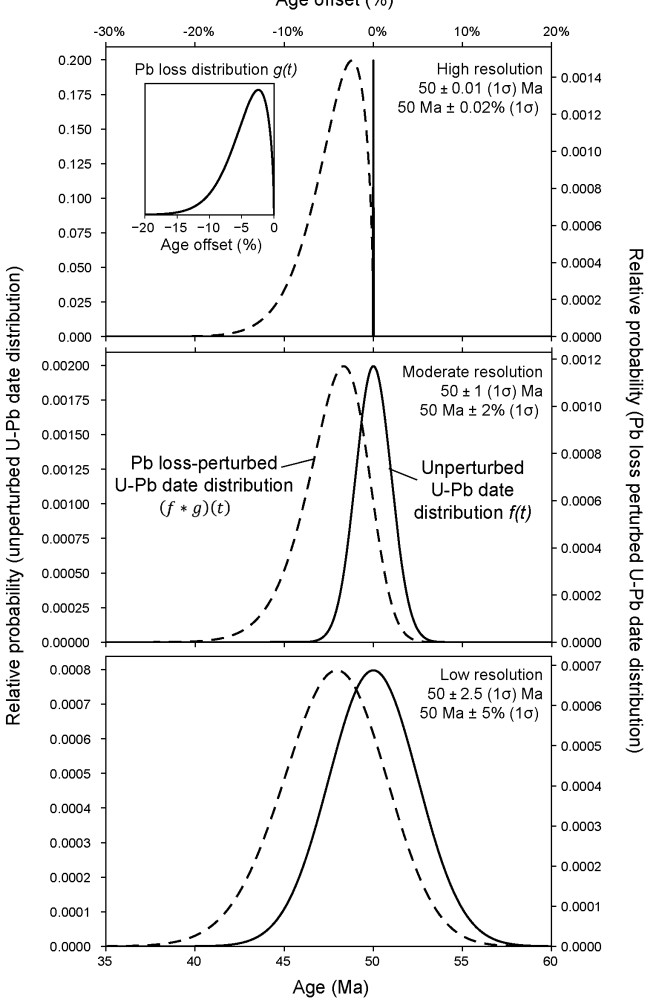

**Figure 8. Illustration of how measurement precision influences the resulting Pb loss-perturbed U-Pb date distribution (dashed black line). The same distribution of Pb loss, *g(t)*, is used for all three panels. The convolution of *f(t)* and *g(t)* closely approximates *g(t)* for high precision analyses whereas the convolution of *f(t)* and *g(t)* has a similar shape to *f(t)* for low precision analyses.**

### 5.5 Strategies for future data collection

Given the limitations of existing samples in modeling apparent Pb loss distributions, we suggest two strategies that might guide future data collection. First, increasing the number of analyses would provide improved characterization of the apparent Pb

loss distribution. In particular, heavy-tailed distributions (e.g., Pareto) predict rare but highly offset values that may not be identified unless a sufficient number of grains are analyzed. We suggest that future work focus on increasing the number of ±CA *in-situ* U-Pb analyses, with ideal datasets having 100s, or even 1000s, of analyses per sample (e.g., Pullen et al., 2014).





Such efforts are likely to be promoted by recent advances in the rate of LA-ICP-MS throughput for improved provenance characterization of detrital zircon (Sundell et al., 2021).


Second, increasing the precision of U-Pb date measurements will facilitate accurate identification of the apparent Pb loss distribution (Fig. 8). As the standard deviation of the unperturbed U-Pb date distribution becomes increasingly small (i.e., due to increasing analytical precision), the convolution of $f(t)$ and $g(t)$ will become increasingly similar to $g(t)$ (Fig. 8). Thus, the same distribution of apparent Pb loss will be more easily discerned for higher precision U-Pb data than for lower precision data. *In-situ* zircon U-Pb dates typically have 1-3% relative uncertainties (Schaltegger et al., 2015; Sharman and Malkowski,

2020), similar to the 'moderate' precision example in Figure 8, whereas the precision achieved by CA-ID-TIMS is commonly better than 0.1% depending on U content and age of the crystal (Schaltegger et al., 2015). Thus, the ideal dataset would consist of large numbers of paired non-CA- and CA-ID-TIMS measurements from randomly selected zircon from the same sample. However, given the practical limitations of cost and time involved with collecting CA-ID-TIMS measurements, we suggest

that ±CA-LA-ICP-MS analysis might offer the best compromise between data quantity (i.e., rapid analysis) and precision.

Figure 8 also illustrates the challenge of accurately identifying the distribution of apparent Pb loss when the degree of Pb loss is small in comparison to the relative precision of the U-Pb dates. For instance, a sample with low amounts of apparent Pb loss (e.g., SRF; Fig. 5) may be equally well modeled by any number of distribution types, as they all are able to collapse

to $W_1$ and $W_2$ values approaching 0. Thus, we anticipate the ability to differentiate between apparent Pb loss distribution types to decrease as the magnitude of apparent Pb loss decreases, except for perhaps the highest resolution U-Pb datasets (e.g., ID-TIMS). However, this limitation may be partly overcome by increasing the number of analyses per sample.

## 6 Conclusions

This study presents a novel framework for quantifying the distribution of apparent Pb loss on U-Pb date distributions, which

could include true loss of radiogenic Pb or other processes that also produce a systematically negative age offset. We show that a Pb loss-perturbed U-Pb date distribution from a set of zircon crystals with a shared crystallization age can be represented by the convolution of the unperturbed Gaussian U-Pb date distribution and the distribution of Pb loss. Our approach relies on analyzing differences between the untreated date distribution from *in-situ* U-Pb geochronology (i.e., LA-ICP-MS or SIMS) and an independent estimate of the true crystallization age, which could include U-Pb dates from a thermally annealed and

chemically abraded aliquot of the same sample or from another geochronometer (e.g., $^{40}Ar/^{39}Ar$). We suggest that the first and second Wasserstein distances ($W_1$ and $W_2$) of the apparent Pb loss distribution can be used to quantify the total degree of apparent Pb loss that a set of grain analyses has undergone, with maximum possible $W_1$ and $W_2$ values of 100.



We apply this modeling framework to ten igneous samples (Miocene to Carboniferous) analyzed with LA-ICP-MS or SIMS.
All but one of the samples showed a high probability that the untreated U-Pb date distribution has been perturbed by Pb loss or other equivalent process. Of the eight types of continuous apparent Pb loss distributions that we considered, the Weibull distribution produced the overall best-fit. However, the number of analyses in the samples we analyzed was generally low (17-144, average of 39), which likely contributed to many of the modeled Pb loss distributions producing statistically acceptable fits (i.e., K-S or Kuiper p-value >0.05). In general, we noted two behaviors of apparent Pb loss in these samples; samples with
a bulk shift in U-Pb date distributions ($W_2/W_1 <\sim 1.3$) and samples where most grains had very little offset but few grains had much larger offsets ($W_2/W_1 >\sim 1.7$). The overall magnitude of apparent Pb loss was also found to be variable, with median values varying from -0.9% to -6.5%.

Although our modeling framework is designed for analysis of cogenetic grains, and thus most appropriate for igneous samples,
we illustrate how the approach could be applied to detrital samples by first unmixing the CA U-Pb date distribution into component Gaussian distributions. However, analysis of detrital samples with zircon of widely varying age and history may violate an assumption of our model that the amount of apparent Pb loss is uncorrelated with age. We thus suggest that multimodal U-Pb date distributions be divided and modeled separately to allow the apparent Pb loss distribution to vary over time.


Given the widespread application of *in-situ* U-Pb geochronology of untreated zircon across many disciplines of geosciences, improved characterization of both the distribution type(s) and magnitude of apparent Pb loss is warranted, particularly for Phanerozoic zircon where cryptic Pb loss is difficult to identify. We highlight a need for increased sampling and high-*n* characterization of paired non-CA and CA *in-situ* U-Pb datasets. ±CA-LA-ICP-MS in particular has potential given the ability
to rapidly acquire the type of large datasets that would facilitate modeling apparent Pb loss distributions. In addition, we recommend simultaneous collection of parameters such as zircon size and trace elemental concentrations to aid in future efforts to understand the mechanisms of negative age offsets. Ultimately, we anticipate that improved characterization of the magnitude and distribution type(s) of apparent Pb loss will aid in interpreting non-CA *in-situ* U-Pb datasets and guide strategies for future data collection.

**Data availability**

Data are archived under https://doi.org/10.5281/zenodo.7783226. Table A1 and Figure A1 include summaries of all model results. Tables A2 and A3 provide U-Pb analytical results for sample MM20-EC-109 from the University of Arizona LaserChron Center (LA-ICP-MS) and Boise State University Isotope Geology Laboratory (CA-ID-TIMS), respectively. Figure A2 includes CL images from the University of Arizona LaserChron Center. Figure A3 provides a summary of all best
fitting continuous Pb loss distributions for each sample.



**Code availability**

Code used in this research is available on GitHub (https://github.com/grsharman/Pb_loss_modeling) with the initial commit archived under https://doi.org/10.5281/zenodo.7783243.


**Video supplement**

Supplemental Video 1 is available at https://doi.org/10.5281/zenodo.7783226. This animation provides an illustration of how a Gaussian distribution of U-Pb dates (solid, blue line), $f(t)$, may be perturbed by exponential Pb loss, $g(t)$ (solid, red line). The exponential Pb loss distribution is first reflected about the y-axis and then iteratively shifted by small values of $t$, $g(t\text{-}\tau)$ (dashed,
red line). The convolution of $f(t)$ and $g(t)$ at any given value of $t$ equals the summed area underneath the product of $f(t)$ and $g(t\text{-}\tau)$.

**Author contribution**

G. Sharman and M. Malkowski co-designed the study. G. Sharman developed the code. M. Malkowski produced the U-Pb
data from sample MM20-EC-109. G. Sharman and M. Malkowski wrote the manuscript.

**Competing interests**

The authors declare that they have no conflict of interest.

**Acknowledgments**

That authors thank Mark Pecha, George Gehrels, and staff at the University of Arizona LaserChron (supported by NSF-EAR awards #1649254 and #2050246) as well as Jim Crowley and Mark Schmitz at the Isotope Geology Laboratory at Boise State University. The project is supported in part by NSF EAR award #2243685, American Chemical Society Petroleum Research Fund award #66408-DNI8, and the industrial affiliate members of the Detrital Geochronological Laboratory. We thank Kevin
Befus for coding advice and Greg Dumond for helpful discussions.

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
