# Peer review of "Modeling apparent Pb loss in zircon U-Pb geochronology"

_Geochronology, 2023_

## Author Comment (AC1)

**Response to Reviewer 1's comments on manuscript gchron-2023-6 "Modeling apparent Pb loss in zircon U-Pb geochronology"**

Glenn R. Sharman and Matthew A. Malkowski
June 7, 2023

Reviewer #1 provided a thoughtful review of our manuscript that highlighted several ways in which our modeling framework could be improved. Below, we explore how we will implement Reviewer #1's suggestions in a future revision, with ***bold, italic text*** highlighting specific changes that we intend to make. We also provide the Reviewer #1's comments below with line numbers, which we reference in our response.

In our original submission, we represented apparent Pb loss as a negative percentage offset from the true crystallization age. Although Reviewer #1 noted that this is mathematically expedient (lines 14-15), they made several worthy suggestions for improving the modeling framework. We see these suggestions as having several inter-connected elements, which we will consider in sequence below.

First, Reviewer #1 pointed out that negative percent offset from the true crystallization age, which is what we modeled, is not equivalent to the percent of Pb lost (line 16). This is a fair point and one that stems from both (1) the non-linear relationship of $^{206}Pb/^{238}U$ and $^{207}Pb/^{235}U$ over time (Fig. 1) and (2) the fact that the relationship between % age offset and % Pb lost is dependent on the timing of when Pb is lost.

[Figure]

Figure 1. The relationship between age and isotopic ratios in the $^{206}Pb/^{238}U$ and $^{207}Pb/^{235}U$ systems.

To understand the first point above, we plotted the relationship between % age decrease and % of Pb loss (at present-day) for zircon crystals of three ages 10, 100, and 1000 Ma for both the $^{206}Pb/^{238}U$ and $^{207}Pb/^{235}U$ systems (Fig. 2). As Reviewer #1 suggested, the relationship is not exactly linear. However, the relationship is very close to linear for the $^{206}Pb/^{238}U$ system and for younger crystals in both systems. For example, if a 500 Ma zircon crystal loses 50% of its Pb at present-day, its $^{206}Pb/^{238}U$ age would decrease by 49.0% and its $^{207}Pb/^{235}U$ age would decrease by 43.9%. This effect is less pronounced for younger crystals; 50% Pb loss in a 100 Ma zircon would produce a 49.8% reduction in $^{206}Pb/^{238}U$ age and a 48.8% reduction in $^{207}Pb/^{235}U$ age. Fig. 2

illustrates that the discrepancy is lower for small or large amounts of Pb loss (i.e., the difference is greatest ~50% Pb loss).

[Figure]

Figure 2. Relationship between relative age decrease and % recent Pb loss.

To address this issue, ***we will change the modeling framework to use isotopic ratios as input data instead of calculated ages***. Figure 3 illustrates that we achieve similar results using either approach for sample 284-2. This is to be expected as the relationship between % age decrease and % Pb loss is approximately linear for young samples and modest amounts of Pb loss in the $^{206}$Pb/$^{238}$U system (Fig. 2). Regardless, we believe that this change will allow the modeling framework to be more accurate and flexible. In our revision, we will ***revise all tables and figures*** after implementing the new modeling approach.

[Figure]

Figure 3. Comparison of modeling apparent Pb loss as a function of $^{206}$Pb/$^{238}$U age (left) versus $^{206}$Pb/$^{238}$U ratio (right) for sample 248-2 (von Quadt et al., 2014). Using isotopic ratios as input into the model yields a similar result.

The second point above relates to the timing of Pb loss (lines 17-26). In our original submission we did not discuss the timing of Pb loss, but rather focused on the magnitude of age offset as a proxy for amount of Pb lost. However, in doing so we failed to clearly articulate an important point that Reviewer #1 raises: % reduction in age (or isotopic ratio) can only be directly related to % Pb loss if the Pb loss event occurred recently.

We explore this idea through a thought experiment of three 100 Ma zircon crystals that experienced Pb loss at different times: 10% after 100 Myr (present-day), 20% after 50 Myr, and 40% after 25 Myr (Fig. 4). All three zircon crystals yield similar $^{206}Pb/^{238}U$ ratios (~0.01406, or ~90 Ma) despite the magnitude of actual Pb loss being very different (Fig. 4). Because the shape of Concordia is nearly linear between 0 and 100 Ma, all three zircon crystals move along similar pathways.

[Figure]

Figure 4. Illustration of how similar $^{206}Pb/^{238}U$ can be obtained by different Pb loss histories. Due to the approximately linear shape of Concordia for young analyses, there is relatively little discordance despite highly contrasting Pb loss histories.

One way of handling this issue in a revision would be to ***better articulate that the % apparent Pb loss reflects the cumulative amount of Pb lost over the history of the zircon crystal, and that this value could be thought of as a 'minimum' because a greater amount of Pb loss in the past would be required to achieve the same reduction in final $^{206}Pb/^{238}U$ ratio*** (Fig. 4). For instance, a 10% reduction in Pb after 100 Myr (present-day) is approximately equivalent to a 20% reduction after 50 Myr in the example provided in Figure 4.

Another way of handling this issue in a revision would be to ***incorporate the timing of Pb loss as an adjustable parameter in the model***. Reviewer #1 specifically asks "Would it be possible to consider instead a convolution between a Gaussian distribution representing the isotopic ratios at the time of Pb-loss and a distribution representing the actual amount of Pb lost?" (lines 22-24). This is indeed possible by adding an additional step: adjusting isotopic ratios backwards in time to the specified timing of the Pb loss event prior to applying the modeling framework to the adjusted data. We have tested this approach and found it to be effective if the timing of the Pb loss event is known. ***We plan on including this flexibility in the revised model.*** Although it was not the goal of the paper to model the timing of Pb loss, there are a number of existing approaches that have been developed for this purpose (including several highlighted by Reviewer #2, e.g., Kirkland et al., 2017). We note that Keller (2023) also recently proposed a Bayesian approach to assessing the timing of Pb loss in *Geochronology Discussions* (https://doi.org/10.5194/gchron-2023-9). Such an approach could be potentially used to constrain the timing of Pb loss as input into our model.

However, we are skeptical that adding the ability to specify the timing of Pb loss will be useful in practice for the samples modeled in this study and for Phanerozoic zircon more generally. This is because it is challenging to determine the timing of Pb loss in "young" (i.e., < several 100 Ma) crystals for several reasons: (1) the shape of the $^{206}Pb/^{238}U$ vs $^{207}Pb/^{235}U$ Concordia line is close to linear and thus discordance as a consequence of Pb loss is minimal, (2) $^{207}Pb/^{235}U$ dates are typically low precision for *in-situ* U-Pb analyses, and (3) amounts of apparent Pb loss measured in the samples we model is relatively low (typically <10%). All of these factors together will likely make it challenging to accurately determine the timing of Pb loss. As briefly discussed in our original submission, these factors also contribute to cryptic Pb loss being challenging to identify, in general, on the basis of comparing the $^{206}Pb/^{238}U$ vs $^{207}Pb/^{235}U$ systems.

Figure 4 illustrates this point; despite vastly different Pb loss histories, all four zircon crystals plot on approximately the same location on the Concordia diagram. Regardless, we will aim for **our revised manuscript to include a discussion of the effects of the timing of Pb loss and allow users the flexibility of modeling % Pb loss in context of when it occurred in the zircon's history.**

This change will also allow Pb loss to be modeled in the $^{207}Pb/^{235}U$ (and theoretically $^{208}Pb/^{232}Th$) systems. In practice we are skeptical that modeling $^{207}Pb/^{235}U$ will be as successful as in the $^{206}Pb/^{238}U$ system due to higher uncertainty in $^{207}Pb/^{235}U$ dates. Regardless, we anticipate that the revised modeling framework will allow the flexibility for modeling all three Pb-based decay chains.

Reviewer #1 makes several additional suggestions, which we summarize below.

1. Consider the effects of a common Pb correction (lines 27-34). ***We will add discussion text that considers the potential influence of common Pb corrections***. See also Reviewer #2s comment. We suspect that a full exploration of this topic may be outside the scope of this manuscript, but we agree that this topic warrants discussion.
2. Directly inverting the Pb loss signal from the data (lines 35-42). This is a good suggestion and one that we have attempted to do. However, we have so far been unsuccessful in directly deconvolving the Pb loss distribution (e.g., we have experimented with scipy.signal functions). This may be in part due to a high degree of noise in our datasets, which are comprised of relatively few analyses. Ultimately, neither of us have expertise in signal processing and would likely need to involve a collaborator. We think that this would be a worthy follow-up manuscript should the approach prove successful.

3. Make it clearer that convolution is essentially adding random variables together (lines 44-50). ***We will add text to clarify that convolution is equivalent to the sum of random variables.*** We intended to communicate this point with the "Z = X + Y" notation at the top of Figure 1. However, clearly this point did not come across in the paper and we can do a better job communicating it. As an aside, we started this project by adding random numbers together, and only realized later than this process can be described by mathematical convolution. So, the suggestion here is well taken.

4. Consider whether "HF leaching sometimes conducted by Ar labs [is comparable to] to CA" in U-Pb. Neither of us have expertise in Ar-Ar geochronology. However, we will ***look into this suggestion and consider adding statements that expand the applicability of the modeling framework, if appropriate***. For example, it's a worthwhile question of whether the U-Pb focused approach that we describe might be exported to other radiogenic systems.

References

Kirkland, C.L., Abello, F., Danišik, M., Gardiner, N.J., and Spencer, C.: Mapping temporal and spatial patterns of zircon U-Pb disturbance: A Yilgarn Craton case study. Gondwana Research 52, 39-47, 2017.

Keller, C.B.: Technical Note: Pb-loss-aware eruption/deposition age estimation. Geochronology Discussions [preprint], https://doi.org/10.5194/gchron-2023-9, in review, 2023.

von Quadt, A., Gallhofer, D., Guillong, M., Peytcheva, I., Waelle, M., and Sakata, S.: U-Pb dating of CA/non-CA treated zircons obtained by LA-ICP-MS and CA-TIMS techniques: impact for their geological interpretation. J. Anal. At. Spectrom 29, 1618-1629, 2014.

The fundamental concept underlying this contribution by Sharman and Malkowski -- that observed
U-Pb ages can be considered as a convolution an a true age distribution (i.e., a distribution
representing analytical uncertainty around the true mean age of the analyzed material) with a
distribution representing Pb-loss -- is certainly reasonable, though the form of these distributions may
vary widely. The analytical age distribution of a single analysis in the absence of Pb-loss is
frequently assumed to be Gaussian, so this is a reasonable assumption; the distribution of Pb-loss is
at present much less well understood. To better understand this latter distribution, the authors start
with independent (arguably Pb-loss-free, CA or non U-Pb) ages for ten Phanerozoic samples, and
convolve each with different potential Pb-loss distributions to see which best reproduces the
observed non-CA U-Pb distribution. While I have a number of questions and suggestions, overall this
is a worthwhile contribution.

The authors represent Pb-loss as a negative percentage offset from the true crystallization age. This is
fine mathematically for the purposes of modelling Pb loss in a single decay system, but perhaps it is
worth emphasizing that

1) this is not equivalent to the percent of Pb lost, and

2) this percentage age difference will not generally be the same for the 206Pb/238U and 207Pb/235U
ages, and for each system will depend on both the time of Pb-loss as well as actual amount of Pb lost

In this context, how do the authors propose to deal with the fact that different "Pb-loss" proportional
age distributions must be convolved for the 206Pb/238U and 207Pb/235U systems? Would it be
possible to consider instead a convolution between a Gaussian distribution representing the isotopic
ratios at the time of Pb-loss and a distribution representing the actual amount of Pb lost? This would
allow the same convolution or deconvolution to apply to both systems simultaneously (and even in
principle 208Pb/232Th).

One other issue arising from the fact that Pb-loss happens in terms of atoms rather than ages is that of
common Pb corrections. In CA-ID-TIMS, common Pb from inclusions is generally thought to be
removed by CA, so only a lab blank subtraction is performed. However, in in-situ analyses some
form of common Pb correction is commonplace; this may have secondary consequences in the case
that a sample is also discordant (e.g., discussion in Andersen et al. 2019, which you currently cite in
the context of the general problem of Pb-loss in in-situ datasets). Fully dealing with this may be
outside the scope of the current paper, but perhaps bears some consideration.

One other conceptual concern involves the form of the distributions chosen to represent Pb-loss; a
number of parametric distributions are tested, and all are better than no correction (with Weibull
performing best), it seems possible that the true distribution of Pb-loss may diverge from any of these
(i.e., be a combination of multiple distributions, or nonparametric). Ideally, it might be possible to
invert for the true form of the Pb-loss distribution.. have the authors considered if a deconvolution /
inverse approach is feasible? Absent that, is there perhaps any underlying quantitative or intuitive
rationale to explain the relative success of the Weibull distribution?

A few other more minor notes:

While the authors do provide several nice illustrations of convolution, one point which may be worth
noting to help make the concept more intuitive to nonspecialists may be that convolving distributions
is equivalent to adding random variables -- so for example convolving an exponential Pb-loss
distribution with a Gaussian analytical distribution yields a third distribution which is the same one
you would draw from by drawing a random variable (i.e., a random age) from the Gaussian and
another from the Exponential and adding them together.

Another point which bears some note: while both CA-ID-TIMS U-Pb ages and Ar/Ar ages are  likely
to avoid the influence of Pb-loss, daughter loss is not unheard of in the Ar/Ar system. How analogous
is the HF leaching sometimes conducted by Ar labs to CA? Is this equally effective in eliminating
daughter loss?

I was glad to see that the authors provided their full code via a persistent DOI (in this case, Zenodo),
in line with best practices. The supplementary video illustrating convolution was a fun addition.

---

## Author Comment (AC2)

**Response to Reviewer 2's comments on manuscript gchron-2023-6 "Modeling apparent Pb loss in zircon U-Pb geochronology"**

Glenn R. Sharman and Matthew A. Malkowski
June 7, 2023

Reviewer #2 provides an overall critical assessment of our manuscript with comments that focused on (1) the novelty of our study, (2) criteria for what types of data should be included in a modeling study, and (3) parts of the text with overstated or imprecise language. Although we generally disagree with Reviewer #2's characterization of points (1) and (2) and provide a rebuttal below, there are many aspects of Reviewer #2's comments that are well taken and that we will incorporate into a future revision. For instance, we now appreciate that there is a need to better describe how our study relates to previously published work on Pb loss modeling in general and to better articulate the specific aims of this study.

Below, we provide a response to Reviewer #2's comments and explain how we will specifically incorporate suggestions in a future revision with ***bold, italic text***. We also provide Reviewer #2's comments below with line numbers, which we reference in our response.

We would like to note that although Reviewer #2 provided extensive comments, this review lacked an assessment of the paper's central thesis – that apparent Pb loss may be characterized by mathematical convolution. Indeed, the review does not use the word "convolution" and does not include comments on the feasibility of the approach – either positive or negative.

**1. Novelty of the study**

Reviewer #2 questions the novelty of our study. For example, Reviewer #2 suggests that "there are already well-established, more appropriate, and more powerful mechanisms" that involve geochemical characterization (U, Fe, Ca, REE, OHO, etc.), analysis via Raman spectroscopy, and internal mineral texture (lines 7-11). Reviewer #2 goes on to provide examples of 'similar population-based approaches' (lines 96-112), citing Morris et al. (2015), Kirkland et al. (2017), Kirkland et al. (2020), and others. Similarly, Reviewer #2 suggests that our treatment of detrital zircon has already been addressed (lines 137-143).

Although we appreciate the suggestions for additional references that relate to the general topic, we would like to emphasize that none of the studies mentioned by Reviewer #2 relate to the specific topic of this paper: assessing distributions of apparent Pb loss magnitude through mathematical convolution. Some of these papers focus on modeling the timing of Pb loss (Morris et al., 2015; Kirkland et al., 2017) and one correcting for common Pb (Anderson, 2002). We thus contest Reviewer #2's assertion that the central thesis of our manuscript has already been published (lines 7-12). None of the references that Reviewer #2 provides mention convolution or attempt to model distributions of apparent Pb loss magnitude. In our future revision, ***we will revisit the introductory text to make sure that we avoid overstating the novelty of this work through specific and precise language about the contribution provided.***

**2. Criteria for a Modeling Study**

Reviewer #2 cites a general community preconception, or skepticism, of model-based studies in that they are "unreliable to the point of being unproductive" (lines 27-32). Reviewer #2 goes on to suggest that new model-based approaches should satisfy two specific set of conditions (lines-27-55). These conditions include specific data types (paired LA-ICP-MS and TIMS, known timing of Pb loss, detailed petrologic data, CL + BSE images, Raman spectroscopy, and mineral chemistry data) (lines 34-55).

Although we don't dispute that the dataset described would likely make an excellent study of Pb loss, we contend that there is no single way to study Pb loss. The type of data collected for a given study should depend on the goals and objective of the study, not a preconditioned list of items specified *a priori*. Reviewer #2 does not explain why the framework of mathematical convolution fails without these data.

There are several reasons why the types of data suggested would be challenging to collect for the purpose of our study. It is not our goal to characterize Pb loss in any single zircon crystal, but rather to characterize the *distribution* of Pb loss magnitude that has influenced an entire sample. This requires ideally many U-Pb analyses from numerous zircon crystals, versus detailed characterization of fewer grains. Specifically, the geochemical data requested (e.g., Fe, Ca, OHO) are not routinely collected during *in-situ* U-Pb geochronology. Routine datasets provide only U and Th concentrations and isotopic ratios ($^{206}Pb/^{238}U$, $^{207}Pb/^{235}U$). We don't dispute that collecting additional data types (e.g., REE) could be useful, and we specifically mentioned this in our original submission (lines 460-462).

This study simply presents a mathematical concept (convolution) for quantifying the distribution of Pb loss magnitude. The datasets we analyze are from, in part, previously published studies that we view in high esteem (e.g., von Quadt et al., 2014; Watts et al., 2016), and the point of our paper mirrors their own: that an offset exists between non-CA and CA U-Pb dates. Our contribution is to provide a mathematical framework for better quantifying that offset in terms of a distribution, rather than as a simple percentage shift. It seems that this simple aim might have been lost on Reviewer #2 (e.g., lines 139-143). In our revision, ***we will revisit the introductory text to make sure that the aim of this paper is clearly communicated.***

Although we disagree with Reviewer #2's criteria for what must be in a modeling study, we do appreciate that the geochronology community is broad and diverse, and that our manuscript might be better received if we were more specific about the context and purpose of this study. In our future revision, ***we will revisit the introductory text to better communicate the aims and context of this paper. We will also better articulate the intended audience of this work (i.e., in-situ U-Pb community) and the expected value. We would also consider changing the title of our paper to be more specific.***

**3. Other Points**

Reviewer #2 highlights several places in the text where our language is vague, overstated or imprecise. Reviewer #2 also highlights opportunities to improve our referencing. We provide a bullet list of these comments and our responses. Overall, these comments will be helpful in improving the revised manuscript.

- Improve the description of causes of radiogenic Pb, specifically by clarifying that fluids are needed to remove Pb (lines 62-65). ***In our revision, we will review the suggested literature and revise the Introductory text appropriately to more completely describe causes of Pb loss.***
- Mention potential causes of non-Gaussian distributions of U-Pb dates, specifically the common Pb correction (see also comment by Reviewer #1) and complexities related to zircon growth (lines 66-84). ***In our revision, we will include improved discussion surrounding our assumption of a starting Gaussian distribution and situations for which this assumption may not be appropriate. We will also clarify that our assumption of Gaussian distribution is one of convenience – mathematical convolution could be done with any distribution type that reflects the underlying non-Pb loss perturbed U-Pb date distribution.***
- Consideration of how the various distributions might relate to geologic processes (lines 113-120). This is a good suggestion, and in our revision ***we will include discussion of potential mechanistic links between geologic processes and distribution types.*** For example, we find it intriguing that the Weibull distribution, which was the best-fitting function for apparent Pb loss distributions, has also been applied to modeling particle size distributions (Zobeck et al., 1999).
- Avoid suggesting that "Pb loss in natural samples has not been well characterized" in the abstract (line 122). In our revision ***we will take care to use more precise language and avoid broad statements***. We agree with the reviewer that much work has been done on open-system behavior in the U-Pb system, and did not intend to imply otherwise.
- Include a more comprehensive list of references (lines 131-133). In our revision, ***we will make more effort to cite relevant studies by a broader diversity of authors, including those mentioned by Reviewer #2***.
- Revise the statement relating to how analyses are pulled off concordia during Pb loss (lines 134-136, referencing line 34 of the original submission). In our revision, ***we will clarify that the analyses may not be pulled completely off of Concordia depending on magnitude of Pb loss and measurement precision, and we will clarify that this takes place during the timing of Pb loss.***
- Mention the fact that open-system behavior is itself useful geologic information that is removed via CA (lines 144-151). This point is well taken. It is not the goal of our work to study the geologic processes associated with Pb loss (e.g., timing). However, this does not mean that this is not useful information for geologic studies. In our revision, ***we will include statements in the discussion that clarify this point***.
- Be more specific about the aim of future data collection (lines 152-161). This is a point well taken. In our revision, ***we will include a better description of the overall goal for which the data collection strategy is oriented***. This goal is to better quantify the distributions of apparent Pb loss magnitude in untreated, *in-situ* LA-ICP-MS analyses.
- Avoid appealing to increasing precision to identifying Pb loss (lines 162-167). We believe that this assertion is valid, as shown by Fig. 8. The complexities mentioned here seem to arise from our assumption of the underlying age distribution being Gaussian. ***We will incorporate additional statements related to this assumption*** (see also lines 66-84 above).

- Consider the implications for thermochronology (lines 168-170). We view this suggestion as being outside of the scope of our study. We are uncertain why the passage of zircon through its closure temperature (He or fission track?) is relevant to the modeling framework that we present.

References

Andersen, T.: Correction of common lead in U-Pb analyses that do not report [204]Pb. Chemical Geology 192, 59-79, 2002.

Kirkland, C.L., Abello, F., Danišik, M., Gardiner, N.J., and Spencer, C.: Mapping temporal and spatial patterns of zircon U-Pb disturbance: A Yilgarn Craton case study. Gondwana Research 52, 39-47, 2017.

Kirkland, C.L., Barnham, M., and Danišik, M.: Find a match with triple-dating: Antarctic sub-ice zircon detritus on the modern shore of Western Australia. Earth and Planetary Science Letters 531, 115953, 2020.

Morris, G.A., Kirkland, C.L., and Pease, V.: Orogenic paleofluid flow recorded by discordant detrital zircons in the Caledonian foreland basin of northern Greenland. Lithosphere 7, 138-143, 2015.

von Quadt, A., Gallhofer, D., Guillong, M., Peytcheva, I., Waelle, M., and Sakata, S.: U-Pb dating of CA/non-CA treated zircons obtained by LA-ICP-MS and CA-TIMS techniques: impact for their geological interpretation. J. Anal. At. Spectrom 29, 1618-1629, 2014.

Watts, K.E., Coble, M.A., Vazquez, J.,A., Henry, C.D., Colgan, J.P., and John, D.A.: Chemical abrasion-SIMS (CA-SIMS) U-Pb dating of zircon from the late Eocene Caetano caldera, Nevada. Chemical Geology 439, 139-151, 2016.

Zobeck, T.M., Gill, T.E., and Popham, T.W.: A two-parameter Weibull function to describe airborne dust particle size distributions. Earth Surface Processes and Landforms 24, 943-955, 1999.

The work by Sharman and Malkowski presents a model-based consideration of the effects of
radiogenic-Pb loss in zircon. Such effects are well known in the U-Pb community and a discussion
on the diagnosis of open system behaviour of widespread importance for U-Pb geochronology.
Nonetheless, there are some significant concerns with aspects of the study that preclude me
recommending publication in its current form.

Specifically, the work apparently seeks to better characterise radiogenic-Pb loss in situations that
it may be cryptic. However, there are already well-established, more appropriate, and more
powerful mechanisms to do this. For example, simple comparison of isotopic ratios to
geochemistry (uranium, iron, calcium, REE, raman, OHO, etc) and / or internal mineral texture
will already provide a much simpler but much more powerful way to demonstrate the presence of

Pb loss. In short, it is unclear how the proposed models provide a tool that will be used to advance
geochronology interpretations.

I am sorry to do this, but I think this work needs to be considered in the historical context of U-Pb
geochronology because it is relevant to perceptions around model-based U-Pb approaches and (as
I get to) has implications for key tests for this work. In the 1960s U-Pb isotopic analyses of zircon
clearly demonstrated that in many cases zircon behaves in an open system fashion (e.g. is
discordant). Now many researchers at that time also attempted to extract primary ages (and
secondary overprinting) by interpreting linear and indeed non-linear arrays on concordia diagrams
using models that rapidly increased in complexity (for example; Tilton 1960 JGR, Silver and
Deutsch 1963 Journal of Geology, Steiger and Wasserburg 1966 JGR). Other developments also
happened at around the time model-based interpretations were in vogue. Namely, isotope dilution
analysis of single zircon grains with air abrasion and magnetic separation (e.g. Krogh) and of
course insitu dating via ion microprobe dating (e.g. Compston). These analytically based
developments set zircon U-Pb geochronology on the pathway of identification, extraction, and
dating of grain domains with closed U-Pb systems (or specific targeting of open system domains
where geochemical evidence could also be brought to bear on the subject).

Now my point (and I am aware of this from my own experience in reviews) the general community
has a strong preconception that model-based approaches are generally unreliable to the point of
being unproductive (given the numerous processes that can lead to the same distribution). Hence,
works that try to revive a model-based approach to U-Pb geochronology, in an effort, to enhance
understanding and make such models helpful to better understand geology, must allay this
perception. In order to achieve this outcome of an advance then what can be done: Well, it would
seem logical to this reviewer, that any new model-based approach needs to satisfy two conditions:

1/ It must be quantitatively calibrated against high quality closed-system geochronological data
AND known times of disturbance. The choice of the samples where both primary and secondary
ages are determined by precise, accurate and model-independent methods for such tests is crucial.
Unfortunately, the sample choice in this work failed this criterion as the same grains were not
analysed after LA-ICPMS by TIMS and in fact, in some cases the choosen studies have used even
a different isotopic system to constrain the "true" age. Moreover, the timing of overprinting
processes has not been clearly independently determined on the same material to the level needed.
Hence, to demonstrate the use of this work and continue this study, such condition really needs to
be passed. Such tests would significantly benefit from including detailed geological and petrologic
information so the geological context and implications of the proposed models can be understood.
This would necessitate detailed characterization of the grains, for example CL and BSE images
before and after analyses, the latter showing ablation spots (and potentially also Raman
spectroscopy) so any relationship between these grain level observations and isotopic ratios could
be made, as they would serve as prima facie evidence of open system conditions.

2) It must be demonstrated that the new approach yields new information that is not available and
unobtainable with modern closed system methods or simple relationships already at hand. This is
a big challenge because by combination of mineral chemistry with isotopic ratios already can yield
much more rigorous insight into geological processes than by this strongly model based example
of age distribution fitting. Furthermore, any ages calculated, or more specifically in this case, distributions proposed with such new methods really needs to be accompanied by uncertainty
intervals that include the model-related uncertainty around the distribution. This is a very difficult
goal to achieve.

In this current study, there appears to be a signficant way to go to satisfactorily address both these
conditions.

**Significant issues**

Precision in the language. There are numerous cases where the level of precision in the text could
lead to miss-interpretation by a reader. Moreover, there are specific inaccuracies. Please refer to
the specific points below which document some of these.

The discussion of the causes of radiogenic Pb loss appears incomplete. While a damaged crystal
structure is clearly a factor it isn't the sole prerequisite for open system processes. Please see the
work of Silver / Pigeon which clearly demonstrates that fluids are also needed to strip Pb. In short,
a more accurate description of radiogenic-Pb loss is needed.

Assumption of a gaussian distribution for the undisturbed zircon state of U-Pb ratios. There are
several primary processes that could lead to a non-gaussian distribution that should at least be
mentioned. While the simplifying assumption of a gaussian distribution is a reasonable starting
position for certain growth processes, the work would be improved with a consideration of the
natural complications to this situation. For example: Common Pb – it's presence and form of
correction. Specifically, a non-uniform common Pb composition (while unlikely to be of
significant concern in zircon and of more relevance for minerals with typically higher common Pb
loads e.g. apatite and titanite) will invalidate the assumption of a gaussian distribution.
Furthermore, there would be expected to be a complex interrelationship between radiogenic-Pb
loss, discordance, and common Pb amount and composition that would have an implication for the
model. Moreover, as precision increases so a natural outcome of this will be a non-gaussian
distribution, the point where this non-gaussian distribution appearance breaks down would be a
function of the growth duration of a population of zircon which is highly magma (size,
temperature, cooling rate, chemistry, etc) dependent. A more sophisticated realisation of what
zircon growth is, would benefit this work (there are several new mineral equilibrium model papers
that deal with zircon growth rates that clearly are relevant in this regard). It is highly simplistic,
without any caveats, to assume zircon growth is instantaneous – there are many environments
where prolonged zircon growth has been demonstrated and these sorts of environments are entirely
unsuited to a model assumption of a normal distribution.

Overlooked published similar population-based approaches in geochronology:

The work makes quite a few claims of novelty. While aspects of the proposed model are indeed
new, there is quite a body of existing work that uses ostensibly, very similar, to similar, to quite
similar approaches to understand: 1/ the most likely timing of radiogenic-Pb loss, 2/ mixing
between different compositional domains and 3/ common Pb correction.

Specifically, the comparison between a model distribution and a measured U-Pb distribution has
in fact been frequently previously utilized and a recognition of this foundation to the present study
clearly required to provide context to this work and demonstrate the advance it makes.

The following works are only those I am aware of, but they may provide some useful context from
which the current model appears developed. It is odd they are not considered and implies some
limitation in the survey of existing literature relevant to this work.

*Pb loss modelling*

1/ Morris et al., 2015, Lithosphere, 138-143; Kirkland et al., 2017, GR, v. 52, 39-47; Kirkland et
al., 2020, GR, v. 77, 223-237. There are probably other publications from this research group that
use distribution comparison techniques to understand Pb loss as well.

Of note here is that the similarity test for the model distribution to the measured distribution is
essentially the same as this work proposes. Surely, this should be acknowledged. The major
difference in these works and the current approach is that they used the observed concordant
distribution in the model whereas the approach proposed in this work is to compare the age
distribution to theoretical distributions.

*Unmixing*

2/ Olierook et al., 2021, GR, v. 92, 102-112.

A similar approach in some regards to address the potential of mixing between different zircon
domains. It also uses a comparison between a reconstructed (e.g. model) distribution and a known
distribution.

*Common Pb correction*

3/ Andersen 2002, CG, v. 192, 59-79.

The common Pb correction approach of Andersen uses some of the same concepts.

The proposed procedure would be able to provide more geological insight if the various
distributions (gamma, Weibull, lognormal, uniform, half normal, pareto etc) compared to the data
were firmly rooted in some dominant geological process. Specifically, the discussion of the
distribution shapes relative to geological processes needs to be significantly enhanced. For
example, even simple end member distributions can be linked to likely geological processes;
radiogenic-Pb loss / uranium gain / Pb gain / U loss, discrete or episodic, common Pb gain,
heterogeneous common Pb, recent Pb loss, ancient Pb loss. In short, more geological context is
required for the patterns that are compared to the measured data.

**Specific points**

Abstract: the authors claim that Pb loss in natural samples has not been well characterized. I would
dispute this, the simplest measure of this process (discordance) is the primary filter essentially
every U-Pb geochronology work uses, there are numerous works considering the process of
radiogenic-Pb loss from the pioneering work of Silver, Pigeon, Krough, Black etc, the field of U-
Pb geochronology has been focused around addressing open system processes (just consider the
formulation of the concordia and Tera-Wasserburg diagrams even). So is it really "not well
characterized"? However, is radiogenic-Pb loss **difficult** to characterise, absolutely it **can** be,
depending on the measurement precision (which itself can be a function of age). This latter aspect
is worth focusing on, to indicate where the proposed modelling approach may have benefits.

Line 26>. Very limited referencing to U-Pb geochronology concepts that appear to favour a
specific author. Suggest providing a more balance and historically accurate list of references that
recognises the contributions to the field.

Line 34. Inaccurate statement, depending on when radiogenic Pb loss has occurred (and the
measurement precision) and the degree of radiogenic Pb loss (e.g. if complete) data may not be off
the concordia curve.

Section 5.3 has specifically been addressed in other works (using a similar more tailored approach)
it seems highly unusual that this context isn't provided here.

Also, the proposed approach for DZ seems incomplete as it is unclear what the purpose of this
modelling is for; is it to better understand the primary crystallization ages, the timing of Pb loss,
or the degree of mixing between different age components in any distribution? Furthermore, the
proposition is somewhat cryptic and certainly difficult to apply to a detrital situation. I really don't
see the contribution this paragraph of text makes to the overall presentation.

A major assumption of this work is that radiogenic-Pb loss is an impediment to understanding. Yet
the reality is that tracking open system processes is possible with radiogenic-Pb loss and depending
on the geological question posed, a very useful way of gaining otherwise difficult to access
geological information. Moreover, the whole point of insitu dating is to characterize the full range
of (texturally / geochemically defined) age components thus providing an understanding of the full
range of geological processes a sample may have undergone. CA work clearly has its place but it
is inevitable that such approach is removing some element of geological information in favour of
another. The text is strongly one sided in its appraisal of CA and its merits or otherwise.

The discussion of strategies for future data collection needs to be very specific about what the aim
of any data collection is; is it to date igneous crystallization, metamorphism, fluid mediated
recrystallization, overprinting thermal events? What? Such fundamental information is necessary
first before the strategy can be evaluated for the proposed purpose because such underlying
geological question would affect everything from required temporal resolution to the most likely
manifestation of radiogenic-Pb loss. Simply arguing for greater number of analyses to better
characterise apparent age distributions seems a rather weak suggestion. The more dominant age
components (be they detrital or caused by radiogenic-Pb loss) will be more likely to be sampled
(assuming random sampling) for any n selected. This aspect appears to be overlooked but the
statistics in some of the DZ work of Anderson and others demonstrate this point.

It is incorrect to appeal to increasing precision alone to identify radiogenic-Pb loss. The natural
extension of this argument ends, rather, with being able to identify the timeframes of which zircon
itself grows; there are plenty of zircon growth models about based on modified equilibrium
pseudosections that demonstrate zircon has variably prolonged growth intervals in certain
environments. Again, the geological environment that the strategy is proposed for needs to be
much better presented (e.g. rapid volcanic crystallization).

Furthermore, it would seem useful to consider the model in the context of thermochronology
considerations where timing through closure temperature is of relevance (e.g. growth within a
magma chamber versus explosive removal from that chamber).

The reality is that strategies should be developed that integrate geochemical parameters of the
zircon to better understand the growth or modification process the U-Pb systematics have been
potentially affected by. Considering the age distribution alone seems a simplistic and potentially
highly misleading approach given the numerous cofounding variables that could give rise to the
same distribution.

---

## Author Response (AR1)

**Department of Geosciences**
Fulbright College of Arts and Sciences

http://geosciences.uark.edu

216 Gearhart Hall
Fayetteville, Arkansas 72701
Office: (479) 575-3355
Fax: (479) 575-3469

August 30, 2023

Dear Editor,

We are re-submitting a manuscript (gchron-2023-6) entitled "**Modeling apparent Pb loss in zircon U-Pb geochronology**" to *Geochronology*. This manuscript provides a novel mathematical framework for deconvolving the influence of Pb loss, or other processes that cause negative Pb*/U offsets, on zircon U-Pb date distributions. We believe that this revised manuscript has been substantially improved following incorporation of feedback by two anonymous reviewers. We have provided both a clean and tracked-changes version of our manuscript. We also provide a document with responses to each reviewer comment. The list below outlines some of the more significant changes that stem from reviewer feedback:

-   Modeling Pb*/U ratios instead of calculated U-Pb ages (Reviewer 1)
-   Clarifying issues related to the timing of Pb loss and incorporating the timing of Pb loss as an adjustable parameter in the model (Reviewer 1)
-   Adding statements that address potential bias from common Pb overcorrection (Reviewers 1 and 2)
-   Clarifying the novelty/aim of this work and putting in better context with previous studies, including additional citations (Reviewer 2)
-   Many additional changes to the text that reflect comments and suggestions from both reviewers

In addition, this revised manuscript incorporates edits that stem from a virtual meeting with associate editor Pieter Vermeesch, including incorporating the logit-normal distribution as the preferred parameterization of *g(t)* and simplifying and shortening the manuscript (e.g., moving Figure 3 to supplemental and removing the discussion related to detrital samples). We have also explored the possibility of modeling Pb loss through the lens of optimal transport modeling, as suggested. Preliminary analysis has shown some promise, but we have reached a conclusion with Alex Lipp that this approach is likely outside of the scope of what can be accomplished in this paper as more analysis is needed.

We believe that this study would be of wide interest to the readership of *Geochronology*, particularly to the *in-situ* U-Pb geochronological community. We have included supporting data, and code is available via a GitHub repository. We thank you for your consideration of our revision.

Sincerely,

Glenn R. Sharman*, Matthew A. Malkowski

* Corresponding author – affiliation: Department of Geosciences, University of Arkansas, Fayetteville, AR, USA, gsharman@uark.edu, mobile: 302-745-1412

The University of Arkansas is an equal opportunity/affirmative action institution.

Responses to Reviewers

Our responses are shown in **red, bold text**

Reviewer 1

The fundamental concept underlying this contribution by Sharman and Malkowski -- that observed U-Pb ages can be considered as a convolution an a true age distribution (i.e., a distribution representing analytical uncertainty around the true mean age of the analyzed material) with a distribution representing Pb-loss -- is certainly reasonable, though the form of these distributions may vary widely. The analytical age distribution of a single analysis in the absence of Pb-loss is frequently assumed to be Gaussian, so this is a reasonable assumption; the distribution of Pb-loss is at present much less well understood. To better understand this latter distribution, the authors start with independent (arguably Pb-loss-free, CA or non U-Pb) ages for ten Phanerozoic samples, and convolve each with different potential Pb-loss distributions to see which best reproduces the observed non-CA U-Pb distribution. While I have a number of questions and suggestions, overall this is a worthwhile contribution.

The authors represent Pb-loss as a negative percentage offset from the true crystallization age. This is fine mathematically for the purposes of modelling Pb loss in a single decay system, but perhaps it is worth emphasizing that

1) this is not equivalent to the percent of Pb lost, and

2) this percentage age difference will not generally be the same for the 206Pb/238U and 207Pb/235U ages, and for each system will depend on both the time of Pb-loss as well as actual amount of Pb lost

In this context, how do the authors propose to deal with the fact that different "Pb-loss" proportional age distributions must be convolved for the 206Pb/238U and 207Pb/235U systems? Would it be possible to consider instead a convolution between a Gaussian distribution representing the isotopic ratios at the time of Pb-loss and a distribution representing the actual amount of Pb lost? This would allow the same convolution or deconvolution to apply to both systems simultaneously (and even in principle 208Pb/232Th).

**Our Response: In our revision we now apply our mathematical framework to Pb\*/U ratios instead of calculated ages. Figures, tables, and the text have been revised accordingly. We now clarify that the modeling approach may be applied to both $^{206}$Pb\*/$^{238}$U and $^{207}$Pb\*/$^{235}$U. We also clarify in the text (Part 3 of Section 5.1) that *g(t)* will underestimate the true magnitude of ancient Pb loss if present-day Pb\*/U ratios are used. We now add the ability in the code to specify the timing of Pb loss as an adjustable parameter.**

One other issue arising from the fact that Pb-loss happens in terms of atoms rather than ages is that of common Pb corrections. In CA-ID-TIMS, common Pb from inclusions is generally thought to be removed by CA, so only a lab blank subtraction is performed. However, in in-situ

analyses some form of common Pb correction is commonplace; this may have secondary consequences in the case that a sample is also discordant (e.g., discussion in Andersen et al. 2019, which you currently cite in the context of the general problem of Pb-loss in in-situ datasets). Fully dealing with this may be outside the scope of the current paper, but perhaps bears some consideration.

**Our Response: We now include a statement relating to common Pb corrections in Part 1 of Section 5.1. Given that this topic has been extensively discussed in other publications (e.g., Anderson et al., 2019), we don't think that a fuller treatment of the topic is necessary in this article. However, we agree that this and other complexities related to the U-Pb system should be considered carefully when applying the modeling framework presented herein.**

One other conceptual concern involves the form of the distributions chosen to represent Pb-loss; a number of parametric distributions are tested, and all are better than no correction (with Weibull performing best), it seems possible that the true distribution of Pb-loss may diverge from any of these (i.e., be a combination of multiple distributions, or nonparametric). Ideally, it might be possible to invert for the true form of the Pb-loss distribution.. have the authors considered if a deconvolution / inverse approach is feasible? Absent that, is there perhaps any underlying quantitative or intuitive rationale to explain the relative success of the Weibull distribution?

**Our Response: Part 2 of Section 5.1 now explicitly mentions some of the benefits and drawbacks of parametric modeling and that our approach should be viewed in the context of exploratory modeling. We agree that development of a nonparametric approach to estimating $g(t)$ would be an improvement. However, nonparametric approaches benefit from high data density (i.e., high-$n$ samples), which currently available samples do not have.**

**We have also greatly simplified the number of parametrizations of $g(t)$ that are presented in the manuscript. We now present just one type of continuous probability distribution, the logit-normal distribution, and explore how this distribution can accommodate different scenarios of Pb loss in Section 3.1. We also include a new figure (Figure 4) that explores this concept.**

A few other more minor notes:

While the authors do provide several nice illustrations of convolution, one point which may be worth noting to help make the concept more intuitive to nonspecialists may be that convolving distributions is equivalent to adding random variables -- so for example convolving an exponential Pb-loss distribution with a Gaussian analytical distribution yields a third distribution which is the same one you would draw from by drawing a random variable (i.e., a random age) from the Gaussian and another from the Exponential and adding them together.

**Our Response: We added a sentence to Section 2 that explicitly makes this point.**

Another point which bears some note: while both CA-ID-TIMS U-Pb ages and Ar/Ar ages are likely to avoid the influence of Pb-loss, daughter loss is not unheard of in the Ar/Ar system. How analogous is the HF leaching sometimes conducted by Ar labs to CA? Is this equally effective in eliminating daughter loss?

**Our Response: Although this paper is focused on the U-Pb system, we see some parallels with loss of daughter product in other geochronological systems. However, we are hesitant to make this connection as neither of us have expertise in the Ar-Ar system.**

I was glad to see that the authors provided their full code via a persistent DOI (in this case, Zenodo), in line with best practices. The supplementary video illustrating convolution was a fun addition.

Reviewer 2

The work by Sharman and Malkowski presents a model-based consideration of the effects of radiogenic-Pb loss in zircon. Such effects are well known in the U-Pb community and a discussion on the diagnosis of open system behaviour of widespread importance for U-Pb geochronology. Nonetheless, there are some significant concerns with aspects of the study that preclude me recommending publication in its current form.

Specifically, the work apparently seeks to better characterise radiogenic-Pb loss in situations that it may be cryptic. However, there are already well-established, more appropriate, and more powerful mechanisms to do this. For example, simple comparison of isotopic ratios to geochemistry (uranium, iron, calcium, REE, raman, OHO, etc) and / or internal mineral texture will already provide a much simpler but much more powerful way to demonstrate the presence of Pb loss. In short, it is unclear how the proposed models provide a tool that will be used to advance geochronology interpretations.

I am sorry to do this, but I think this work needs to be considered in the historical context of U-Pb geochronology because it is relevant to perceptions around model-based U-Pb approaches and (as I get to) has implications for key tests for this work. In the 1960s U-Pb isotopic analyses of zircon clearly demonstrated that in many cases zircon behaves in an open system fashion (e.g. is discordant). Now many researchers at that time also attempted to extract primary ages (and secondary overprinting) by interpreting linear and indeed non-linear arrays on concordia diagrams using models that rapidly increased in complexity (for example; Tilton 1960 JGR, Silver and Deutsch 1963 Journal of Geology, Steiger and Wasserburg 1966 JGR). Other developments also happened at around the time model-based interpretations were in vogue. Namely, isotope dilution analysis of single zircon grains with air abrasion and magnetic separation (e.g. Krogh) and of course insitu dating via ion microprobe dating (e.g. Compston). These analytically based developments set zircon U-Pb geochronology on the pathway of identification, extraction, and dating of grain domains with closed U-Pb systems (or specific targeting of open system domains where geochemical evidence could also be brought to bear on the subject).

**Our Response: We now include additional citations to key sources. We also include additional statements in the Introduction that clarify the relationship of this work to previous study on the general topic of Pb loss.**

Now my point (and I am aware of this from my own experience in reviews) the general community has a strong preconception that model-based approaches are generally unreliable to the point of being unproductive (given the numerous processes that can lead to the same distribution). Hence, works that try to revive a model-based approach to U-Pb geochronology, in an effort, to enhance understanding and make such models helpful to better understand geology, must allay this perception. In order to achieve this outcome of an advance then what can be done: Well, it would seem logical to this reviewer, that any new model-based approach needs to satisfy two conditions:
1/ It must be quantitatively calibrated against high quality closed-system geochronological data AND known times of disturbance. The choice of the samples where both primary and secondary ages are determined by precise, accurate and model-independent methods for such tests is crucial. Unfortunately, the sample choice in this work failed this criterion as the same grains were not analysed after LA-ICPMS by TIMS and in fact, in some cases the choosen studies have used even a different isotopic system to constrain the "true" age. Moreover, the timing of overprinting processes has not been clearly independently determined on the same material to the level needed. Hence, to demonstrate the use of this work and continue this study, such condition really needs to be passed. Such tests would significantly benefit from including detailed geological and petrologic information so the geological context and implications of the proposed models can be understood. This would necessitate detailed characterization of the grains, for example CL and BSE images before and after analyses, the latter showing ablation spots (and potentially also Raman spectroscopy) so any relationship between these grain level observations and isotopic ratios could be made, as they would serve as prima facie evidence of open system conditions.
2) It must be demonstrated that the new approach yields new information that is not available and unobtainable with modern closed system methods or simple relationships already at hand. This is a big challenge because by combination of mineral chemistry with isotopic ratios already can yield much more rigorous insight into geological processes than by this strongly model based example of age distribution fitting. Furthermore, any ages calculated, or more specifically in this case, distributions proposed with such new methods really needs to be accompanied by uncertainty intervals that include the model-related uncertainty around the distribution. This is a very difficult goal to achieve.
In this current study, there appears to be a signficant way to go to satisfactorily address both these conditions.

**Our Response: Please refer to Point #2 in our "Response to Reviewer 2's comments on manuscript gchron-2023-6" dated June 7, 2023.**

Significant issues
Precision in the language. There are numerous cases where the level of precision in the text could lead to miss-interpretation by a reader. Moreover, there are specific inaccuracies. Please refer to the specific points below which document some of these.

The discussion of the causes of radiogenic Pb loss appears incomplete. While a damaged crystal structure is clearly a factor it isn't the sole prerequisite for open system processes. Please see the work of Silver / Pigeon which clearly demonstrates that fluids are also needed to strip Pb. In short, a more accurate description of radiogenic-Pb loss is needed.

**Our Response: We have added a citation to Pidgeon et al. (1966) and added "exposure to hydrothermal alteration" as a mechanism of Pb loss in the Introduction.**

Assumption of a gaussian distribution for the undisturbed zircon state of U-Pb ratios. There are several primary processes that could lead to a non-gaussian distribution that should at least be mentioned. While the simplifying assumption of a gaussian distribution is a reasonable starting position for certain growth processes, the work would be improved with a consideration of the natural complications to this situation. For example: Common Pb – it's presence and form of correction. Specifically, a non-uniform common Pb composition (while unlikely to be of significant concern in zircon and of more relevance for minerals with typically higher common Pb loads e.g. apatite and titanite) will invalidate the assumption of a gaussian distribution. Furthermore, there would be expected to be a complex interrelationship between radiogenic-Pb loss, discordance, and common Pb amount and composition that would have an implication for the model. Moreover, as precision increases so a natural outcome of this will be a non-gaussian distribution, the point where this non-gaussian distribution appearance breaks down would be a function of the growth duration of a population of zircon which is highly magma (size, temperature, cooling rate, chemistry, etc) dependent. A more sophisticated realisation of what zircon growth is, would benefit this work (there are several new mineral equilibrium model papers that deal with zircon growth rates that clearly are relevant in this regard). It is highly simplistic, without any caveats, to assume zircon growth is instantaneous – there are many environments where prolonged zircon growth has been demonstrated and these sorts of environments are entirely unsuited to a model assumption of a normal distribution.

**Our Response: We have revised the text to better explain that our assumption of a Gaussian distribution relates to random variability in repeated measurements about the true isotopic value. To support this, we now cite Schoene et al. (2013) who state: "…random uncertainties vary in an unpredictable manner, usually with an assumed Gaussian distribution, and include analytical uncertainties in isotope ratio mass spectrometry." We have rewritten Part 4 of Section 5.1 to further clarify that our approach assumes no geologic variation in the true crystallization age, which is a simplification as even autocrystic zircon crystallize over $10^3$-$10^4$ yr timescales.**

**Complexities related to common Pb corrections are now considered in Part 1 of Section 5.1.**

Overlooked published similar population-based approaches in geochronology:
The work makes quite a few claims of novelty. While aspects of the proposed model are indeed new, there is quite a body of existing work that uses ostensibly, very similar, to similar, to quite similar approaches to understand: 1/ the most likely timing of radiogenic-Pb loss, 2/ mixing between different compositional domains and 3/ common Pb correction.

Specifically, the comparison between a model distribution and a measured U-Pb distribution has in fact been frequently previously utilized and a recognition of this foundation to the present study clearly required to provide context to this work and demonstrate the advance it makes. The following works are only those I am aware of, but they may provide some useful context from which the current model appears developed. It is odd they are not considered and implies some limitation in the survey of existing literature relevant to this work.

Pb loss modelling

1/ Morris et al., 2015, Lithosphere, 138-143; Kirkland et al., 2017, GR, v. 52, 39-47; Kirkland et al., 2020, GR, v. 77, 223-237. There are probably other publications from this research group that use distribution comparison techniques to understand Pb loss as well.

Of note here is that the similarity test for the model distribution to the measured distribution is essentially the same as this work proposes. Surely, this should be acknowledged. The major difference in these works and the current approach is that they used the observed concordant distribution in the model whereas the approach proposed in this work is to compare the age distribution to theoretical distributions.

Unmixing

2/ Olierook et al., 2021, GR, v. 92, 102-112.

A similar approach in some regards to address the potential of mixing between different zircon domains. It also uses a comparison between a reconstructed (e.g. model) distribution and a known distribution.

Common Pb correction

3/ Andersen 2002, CG, v. 192, 59-79.

The common Pb correction approach of Andersen uses some of the same concepts.

The proposed procedure would be able to provide more geological insight if the various distributions (gamma, Weibull, lognormal, uniform, half normal, pareto etc) compared to the data were firmly rooted in some dominant geological process. Specifically, the discussion of the distribution shapes relative to geological processes needs to be significantly enhanced. For example, even simple end member distributions can be linked to likely geological processes; radiogenic-Pb loss / uranium gain / Pb gain / U loss, discrete or episodic, common Pb gain, heterogeneous common Pb, recent Pb loss, ancient Pb loss. In short, more geological context is required for the patterns that are compared to the measured data.

**Our Response: We now cite several of the suggested references in the revised manuscript. However, we contend that the relevance of some of these references to our study is overstated (Point #1 in our "Response to Reviewer 2's comments on manuscript gchron-2023-6" dated June 7, 2023.). For example, although Olierook et al. (2021) provide a useful approach for analyzing rim-core mixtures, this approach uses an unmixing paradigm which is distinct from the approach of mathematical convolution used in our article. Thus, while the approach used by Olierook et al. (2021) permits analysis of individual U-Pb dates (i.e., estimation of a core age with uncertainty), our approach is not applicable to individual U-Pb dates (i.e., we are unable to resolve the amount of Pb loss any given analysis has experienced) and instead relies upon analysis of Pb\*/U distributions.**

Specific points

Abstract: the authors claim that Pb loss in natural samples has not been well characterized. I would dispute this, the simplest measure of this process (discordance) is the primary filter essentially every U-Pb geochronology work uses, there are numerous works considering the process of radiogenic-Pb loss from the pioneering work of Silver, Pigeon, Krough, Black etc, the field of U-Pb geochronology has been focused around addressing open system processes (just consider the formulation of the concordia and Tera-Wasserburg diagrams even). So is it really "not well characterized"? However, is radiogenic-Pb loss difficult to characterise, absolutely it can be, depending on the measurement precision (which itself can be a function of age). This latter aspect is worth focusing on, to indicate where the proposed modelling approach may have benefits.

**Our Response: We now specify that we are referring to "Pb loss distributions", versus "Pb loss" more generally. Although there has been a long history of study of open system behavior in the U-Pb system, this article is narrowly focused on a method for estimating apparent Pb loss *distributions*, which we contend is a novel contribution.**

Line 26>. Very limited referencing to U-Pb geochronology concepts that appear to favour a specific author. Suggest providing a more balance and historically accurate list of references that recognises the contributions to the field.

**Our Response: We now cite Davis et al. (2003), which includes a discussion of the history of U-Pb geochronology in zircon. We have removed two Schoene references to avoid over-citing any given author.**

Line 34. Inaccurate statement, depending on when radiogenic Pb loss has occurred (and the measurement precision) and the degree of radiogenic Pb loss (e.g. if complete) data may not be off the concordia curve.

**Our Response: We have deleted this sentence.**

Section 5.3 has specifically been addressed in other works (using a similar more tailored approach) it seems highly unusual that this context isn't provided here. Also, the proposed approach for DZ seems incomplete as it is unclear what the purpose of this modelling is for; is it to better understand the primary crystallization ages, the timing of Pb loss, or the degree of mixing between different age components in any distribution? Furthermore, the proposition is somewhat cryptic and certainly difficult to apply to a detrital situation. I really don't see the contribution this paragraph of text makes to the overall presentation.

**Our Response: We have removed the section relating to modeling detrital samples in the revised manuscript.**

A major assumption of this work is that radiogenic-Pb loss is an impediment to understanding. Yet the reality is that tracking open system processes is possible with radiogenic-Pb loss and depending on the geological question posed, a very useful way of gaining otherwise difficult to access geological information. Moreover, the whole point of insitu dating is to characterize the full range of (texturally / geochemically defined) age components thus providing an

understanding of the full range of geological processes a sample may have undergone. CA work clearly has its place but it is inevitable that such approach is removing some element of geological information in favour of another. The text is strongly one sided in its appraisal of CA and its merits or otherwise.

**Our Response: We appreciate the perspective provided here that 'noise' to someone might be the 'signal' to someone else. We now include a statement in the Introduction that mentions the geologic information that Pb loss events can provide (citing Morris et al., 2015 and Kirkland et al., 2017). Even though the motivation for this work stems from the issue of incorrectly interpreting crystallization ages from Pb loss perturbed Pb\*/U dates, particularly for Mesozoic and younger zircon, the approach used provides quantification of Pb loss distributions which could be of use to those who are interested in the Pb loss event itself.**

The discussion of strategies for future data collection needs to be very specific about what the aim of any data collection is; is it to date igneous crystallization, metamorphism, fluid mediated recrystallization, overprinting thermal events? What? Such fundamental information is necessary first before the strategy can be evaluated for the proposed purpose because such underlying geological question would affect everything from required temporal resolution to the most likely manifestation of radiogenic-Pb loss. Simply arguing for greater number of analyses to better characterise apparent age distributions seems a rather weak suggestion. The more dominant age components (be they detrital or caused by radiogenic-Pb loss) will be more likely to be sampled (assuming random sampling) for any n selected. This aspect appears to be overlooked but the statistics in some of the DZ work of Anderson and others demonstrate this point.

**Our Response: We have deleted the sections 'Detrital and other multi-modal samples' and 'Strategies for future data collection'.**

It is incorrect to appeal to increasing precision alone to identify radiogenic-Pb loss. The natural extension of this argument ends, rather, with being able to identify the timeframes of which zircon itself grows; there are plenty of zircon growth models about based on modified equilibrium pseudosections that demonstrate zircon has variably prolonged growth intervals in certain environments. Again, the geological environment that the strategy is proposed for needs to be much better presented (e.g. rapid volcanic crystallization).

**Our Response: Part 4 of Section 5.1 now includes statements related to timescales of zircon growth in magmatic systems**

Furthermore, it would seem useful to consider the model in the context of thermochronology considerations where timing through closure temperature is of relevance (e.g. growth within a magma chamber versus explosive removal from that chamber).
The reality is that strategies should be developed that integrate geochemical parameters of the zircon to better understand the growth or modification process the U-Pb systematics have been potentially affected by. Considering the age distribution alone seems a simplistic and potentially highly misleading approach given the numerous cofounding variables that could give rise to the same distribution.

**Our Response: We believe that considerations that relate to thermochronology are likely outside of the scope of this manuscript, which has a very specific focus and aim.**

---

## Editor Decision (ED1)

Dear Dr. Sharman and Dr. Malkowski,

I would like to thank you for submitting your work to *Geochronology*. Your manuscript entitled "Modeling apparent Pb loss in zircon U-Pb geochronology" discusses an important subject that is appropriate for the journal. Your proposed solution (deconvolution of crystallisation and Pb-loss distributions) holds promise. However, the reviewers have identified some fundamental issues with this solution, which make the paper unsuitable for publication in its present form. Your response to the reviewers addresses some of their concerns, but not all of them.

Reviewer 1 notes that your method uses dates, not atoms. It (intentionally?) ignores the physics of Pb-loss. Your paper does not include a single concordia diagram, reflecting the fact that these are not part of the algorithm. Reviewer 1 points out that the $^{238}$U-$^{206}$Pb and $^{235}$U-$^{207}$Pb systems respond differently to Pb loss. Your response to this comment comprises two parts. First, you propose to apply the deconvolution algorithm to the $^{206}$Pb/$^{238}$U ratios instead of the ages. Second, you argue that $^{207}$Pb/$^{235}$U ratios should be ignored because they cannot be measured with the same level of precision as the $^{206}$Pb/$^{238}$U ratios.

Reviewer 2 raises an important concern, which you did not address in your response: "numerous processes [...] can lead to the same distribution".

The implications of both comments on your algorithm can be illustrated with the following three examples:

[Figure]

These three synthetic samples have a very different geological significance, but are characterised by identical $^{206}$Pb/$^{238}$U ratio (and, hence, date) distributions. Plugging them into your deconvolution algorithm will produce three identical solutions.

The example in panel b) could be dismissed as a manifestation of the "garbage in, garbage out" phenomenon. However, the example of panel c) is more troubling. Subjecting this concordant dataset (containing a detrital or xenocrystic age signature, say) to the deconvolution algorithm will produce a mixture of two meaningless distributions ("good data in, garbage out").

As a basic sanity check, one would expect that, in the absence of Pb-loss, the deconvolution algorithm should yield the raw age distribution. Your method does not pass this sanity check. It would be dangerous to release such an algorithm into the wild.

Another aspect of Reviewer 2's comment is that the solutions produced by your deconvolution algorithm are *non-unique*. The paper claims that the Weibull distribution best describes the Pb-loss distribution, but does not offer any explanation why this is the case. Your algorithm tries 11 distributions. Why stop there? The space of probability distributions is infinite. For each dataset, there is a frequency distribution that completely describe it. In fact, there are infinitely many of them. In its present form, there does not appear to be any mechanism in your algorithm to constrain the functional form of the proposal distributions, or the number of parameters needed to describe them. This lack of theoretical justification further reduces the scientific interpretability of the algorithm's output.

In light of these issues, I am unfortunately unable to accept your paper for publication in *Geochronology*. However, should you find a way to address the above concerns, then I would be happy to reconsider a suitably revised version of the paper in the future.

Please do not hesitate to contact me directly should you have any questions about this decision.

With kind regards,

Pieter Vermeesch

---

## Author Response (AR2)

Responses to Reviewer 2 (R1)

I am very pleased to say the authors have taken a generally positive approach to the revisions of the work and as a result, I congratulate them on a more balanced, targeted, informative, and ultimately useful contribution. Well-done! I provide a few more suggestions, which the authors may find of use. I look forward to seeing the final work published.

Some of the claims in the revision about balancing out references haven't been done, again I point out to the authors that in the first paragraph there is three different references to a single author with respect to rather generic TIMS approaches / applications. I don't think this is balanced on this topic for the concepts that are referenced.

**Our Response: We have deleted the Schoene et al. (2010) and Schoene et al. (2012) references to avoid over-citing any single author.**

It is still unfortunate that all of the chosen examples don't have independently constrained times of radiogenic-Pb loss.
Line 6. Suggest rewording so as not to start with "Because" and for precision in meaning consider the following suggestion, e.g. "The loss of radiogenic Pb from zircon is known to be a major factor that can cause inaccuracy in determining primary ages from the U-Pb geochronological system, hence there is a need to better characterize the distribution of Pb loss in natural samples".

**Our Response: We have made this change**

The authors may prefer to us en dash for U–Pb as I believe this is technically the correct use.

**Our Response: We now use the longer dash in all occurrences of "U–Pb"**

Fig 5 is particularly nice.
Line 262 "Common Pb corrections, particularly the 207 Pb-correction, may also introduce a bias towards artificially low Pb*/U values (Anderson, 2002; Anderson et al., 2019)". I don't believe this is correct as it is feasible that the common component in the 207 correction could be too radiogenic (e.g. to high common 7/6) the implication of this would be a distribution tail towards a bias towards lower 238/206, conversely a lower common 7/6 would result in a higher (younger) 207 corrected 8/6 age. The point is that bias can be either way.

**Our Response: We have rephased the sentence "…may introduce a bias in Pb*/U values." This avoids suggesting that the 207 Pb-correction bias is exclusively towards younger 206Pb*/238U values.**

Lines 43-49; I am not sure if this paragraph is of much use because unless some alpha dose calculations or raman measurements are done it appears very unconstrained. In the end, greater Pb loss could be a function of grain size, chemistry, fluids, structure, thermal history etc. All are factors that most people will be well aware of.

**Our Response: We have rewritten the paragraph to emphasize the lack of correlation between apparent Pb loss and alpha dose, instead of referencing age alone. We have also clarified the importance of metamictization as a mechanism of Pb loss in paragraph 2 of the Introduction and also included alpha dose values in Table 1.**